# Cognitive & motor skill transfer across speeds: A video game study

**Pierre Giovanni Gianferrara** *, Shawn Betts, John Robert Anderson

Department of Psychology, Carnegie Mellon University, Pittsburgh, Pennsylvania, United States of America

* pgianfer@andrew.cmu.edu

## Abstract

We examined the detailed behavioral characteristics of transfer of skill and the ability of the adaptive control of thought rational (ACT-R) architecture to account for this with its new Controller module. We employed a simple action video game called *Auto Orbit* and investigated the control tuning of timing skills across speed perturbations of the environment. In *Auto Orbit*, players needed to learn to alternate turn and shot actions to blow and burst balloons under time constraints imposed by balloon resets and deflations. Cognitive and motor skill transfer was assessed both in terms of game performance and in terms of the details of their motor actions. We found that skill transfer across speeds necessitated the recalibration of action timing skills. In addition, we found that acquiring skill in *Auto Orbit* involved a progressive decrease in variability of behavior. Finally, we found that players with higher skill levels tended to be less variable in terms of action chunking and action timing. These findings further shed light on the complex cognitive and motor mechanisms of skill transfer across speeds in complex task environments.

**Data Availability Statement:** All data and software can be found on the KiltHub public repository. Data are located here: https://doi.org/10.1184/R1/13834643 and software can be accessed here: https://doi.org/10.1184/R1/13836224.

## Introduction

The common saying is that "*practice makes perfect*". Expert musicians, chess players, cooks and sports players have all acquired and perfected their skills through hours spent practicing and training [1,2]. Although defined in a variety of ways, the notion of "skill" is often understood as referring to an individual's growing ability to achieve a goal within a specific domain through practice [3]. In the context of skill acquisition, researchers often distinguish between lower-skilled and higher-skilled individuals based on objective quantifiable measures of performance [4]. Snoddy [5] was one of the first to investigate the process of skill acquisition in a mirror-tracing perceptual-motor experiment. The researcher showed that acquiring a skill involved fast improvements early on, which progressively slowed down as performance asymptotically approached a learning plateau. One common way of describing this learning trajectory is to express it in terms of a power function regardless of the task at hand [6] (but see [7–9] for further discussion on the power function).

The process of acquiring a skill may be characterized by the progressive shift from *declarative* knowledge about a specific skill domain, based on verbal instructions, to *procedural* knowledge resulting from the transfer of facts into specific procedures on how to achieve a

**Funding:** This research was supported by the Office of Naval Research (https://www.onr.navy.mil/) Grant N00014-15-1-2151 to JRA, and by Air Force Research Laboratory (AFOSR/AFRL - https://www.afrl.af.mil/) award FA9550-18-1-0251 to JRA. The funders had no role in the study design, data collection and analysis, decision to publish, or preparation of the manuscript.

**Competing interests:** The authors have declared that no competing interests exist.

goal [10–12]. According to Fitts & Posner [13], this progressive shift can roughly be decomposed into three discrete stages of skill acquisition consisting of a cognitive stage where the learner needs to understand how they may achieve the task goal, an associative stage in which the learner relies on feedback to refine their movements and strategy, and an autonomous stage in which motor movements become automatic and require a lower level of attentional capacity [13–15].

Over the past century, experimental psychologists have designed a variety of tasks spanning different domains to further unravel the mechanisms of skill acquisition. Such tasks have typically ranged from low-level perceptual-motor tasks with non-symbolic goals such as Snoddy [5]'s mirror-tracing paradigm [3,5,16,17] to high-level cognitive tasks with symbolic goals and a stronger intellectual component, such as problem solving in mathematics [3,10]. As part of this spectrum, researchers have traditionally characterized motor tasks in terms of reaction times and basic motor control processes [4,16,17]. However, later research resulting from the cognitive revolution challenged this perspective and started emphasizing the additional involvement of cognitive processes such as chunking or feedback learning in motor control [18–20]. Further work by Ackerman [21,22] provided evidence from psychomotor experiments suggesting that early cognitive stages of skill acquisition required general intelligence and attention skills whereas later procedural stages relied more on one's perceptual and motor abilities.

From a motor learning perspective, skill acquisition may be defined in terms of the neuronal changes that enable an organism to execute a motor task better, faster, and more accurately over time [23,24]. To aid with skill learning, one commonly relies on feedback and error detection as a way to appropriately adapt one's motor behavior in a given task. While the process of learning from feedback was originally characterized cognitively in terms of a self-regulating closed-loop system [19] and generalized motor programs [4,20], recent accounts have instead characterized feedback learning from a biological perspective within the framework of optimal feedback control with less emphasis on cognition [25,26]. Optimal control theories typically assume that biological systems learn to produce motor commands by optimizing their behavior with respect to well-specified goals, thus minimizing task-relevant motor variability [25,26]. Other computational accounts of motor adaptation have posited that the brain may compute a sensory prediction error which corresponds to the mismatch between a predicted sensory state prior to performing a movement and the actual state of the system after performing that movement [27–29]. Through a cerebellar-mediated supervised learning mechanism, the brain may refine motor skills with practice by progressively minimizing the sensory prediction error signal, thus optimizing motor behavior [30,31].

Motor skill learning may be facilitated by a range of cognitive and motor factors. One common cognitive strategy is to group motor actions into higher-level *chunks* over the course of practice [18,32–34]. Chunking has often been studied as part of motor sequence learning experiments, such as the serial reaction time task in which participants attend to series of visual targets and quickly respond by pressing a corresponding key. In such studies, the common finding is that reaction time progressively decreases over time as the participant is acquiring skill [33,35,36]. From a cognitive perspective, action chunking provides an advantage in terms of performance in so far as chunks can be retrieved faster, they can be executed more fluently, and they lead to fewer performance errors than individual action plans [24,37,38].

At a motor level, one important factor of skill learning is the timing of motor actions. In the literature, music performance studies have traditionally operationalized motor timing variability in terms of the coefficient of variation (CV), which is the standard deviation of the inter-tap intervals over their mean [39–42]. In such studies, CV is often defined as a measure of tapping consistency independent of the relationship between taps and musical events (e.g., see [42]),

and has previously been used to assess participants' learning of the right tempo when asked to tap along with a musical excerpt [43]. The classic finding is that CV decreases with practice as motor skill improves [39].

However, other experimental measures of motor timing exist in the literature. For instance, early music information retrieval studies were originally concerned with retrieving key parameters of a musical excerpt based on a pianist's interpretation at a keyboard [44,45]. Based on Musical Instrument Digital Interface (MIDI) files which specify keypress onset and offset timestamps, it is possible to extract metrical information about a musical piece such as its rhythmic beat by utilizing dynamic programming algorithms [46,47] or the autocorrelation function of keypress on- and offsets [48]. In the case of the autocorrelation function, metrical beat extraction usually involves identifying the peak with highest amplitude whose periodicity indicates the performer's tempo during performance. One advantage of these methods is that they may reverse engineer the key cognitive parameters related to a musician's performance.

One way to further understand skill acquisition is to study people's ability to transfer their acquired skills to new situations [49]. Not only do skill transfer investigations provide clues as to how skills learned in one setting may be transferred to another setting, they also act as useful experimental tests of existing learning theories which must be applicable across different settings [49,50]. Thorndike [51] was one of the first experimental researchers who theorized about skill transfer. According to Thorndike, skill transfer across two different tasks happens as the result of knowledge elements being identical [3,51,52]. In their investigation of text editing and problem solving in mathematics, Singley & Anderson [11,49] refined Thorndike's *identical elements* theory of transfer and argued that the elements of transfer could be represented in terms of production rules in the adaptive control of thought (ACT) cognitive architecture. More recently, Taatgen [53] proposed the *primitive elements theory of cognitive skills* which decomposes production rules into a number of primitive information processing elements that can be thought of as "sub"-actions related to different working memory components (e.g., perceive an object, press a key, recall a fact from declarative memory). According to Taatgen [53], procedural knowledge can be decomposed into a fixed set of general strategies, which may be innate [10], and a subset of learned task-specific strategies, which may be transferred across tasks that involve overlapping production rules [53].

In this study, we explore issues of cognitive and motor skill transfer in the realm of video games. The use of video game paradigms in the context of experimental psychology, referred to as *Game-XP* [54], has proved to be a useful way of uncovering the basic processes of cognition, perception, and dynamic decision-making in skill acquisition [54]. This perspective is consistent with Allen Newell's recommendation that we study tasks that are a "genuine slab of human behavior" [55]. In this regard, cognitive architectures such as Anderson's adaptive control of thought rational (ACT-R) architecture [56,57] are useful as a way to understand how perceptual, cognitive, and motor mechanisms may be integrated in complex task environments such as action video games [57]. Arguably, the video game that has been most extensively researched in the experimental psychology literature is *Space Fortress* [57–59]. In this game, the player navigates a spaceship in a frictionless environment and needs to fire missiles at a fortress while avoiding getting hit by shells.

One recent addition to ACT-R is the Controller module, which was introduced by Anderson *et al*. [57] in a modeling study involving variants of the *Space Fortress* video game. The purpose of the Controller is to model the experience-dependent process whereby skills stored in individual trackers are progressively tuned through trial and error based on features of the environment that provide feedback. The process itself is referred to by the authors as *control tuning* and is mainly composed of three steps: First, the Controller selects a range of possible parameter values that control an action. This initial range is hypothesized to be roughly

estimated by subjects, who often have a sense of what good values are. Second, the module attends to relevant feedback (specified environmental features), which drives the learning of the appropriate parameter range. Third, the Controller progressively converges towards a selected value for that parameter. Mathematically, this convergence process is governed by the progressive narrowing of a quadratic function, and the probabilistic selection of parameters based on a softmax equation.

Though control tuning mechanisms share some similarity with instance-based learning methods [60–62], as introduced in past ACT-R modeling work on prospective time interval estimation [63–65], they also differ from instance-based learning in important ways that are worth noting. Instance-based learning theories typically assume that increases in performance accuracy levels are due to retrieval across a growing pool of experiences stored in declarative memory [60]. Control tuning does not assume that experiences are stored in declarative memory, and thus does not rely on retrieval strategies. Rather, the Controller is more similar to the notion of "internal model", which has been developed in the motor learning literature [26,30,66], and refers to unconscious mapping of controllable movement properties (e.g., timing, force, direction) to features of the movement (e.g., state, position, velocity) [57]. In video games, one must simultaneously learn a range of sensorimotor skills at very fast speeds (often $< 1$ s), which are all required for the successful completion of the goal. Though instance-based learning can certainly simulate the learning and adaptability of one such skill (e.g., time interval estimation [63–65]), it unfortunately falls short when the skills at hand involve a stronger motor component and when the number of sensorimotor skills to learn is too large to keep track of with conscious control from working memory [26]. In terms of computational efficiency, retrieving chunks from declarative memory is costly and often too slow for sensorimotor skills that need to be rapidly and precisely executed [57]. In this sense, control tuning is a useful addition to the ACT-R architecture which nicely complements production compilation to efficiently simulate skill acquisition in complex tasks [57].

The main research goal of this study is to integrate measures of performance and motor learning to determine whether control tuning as specified by Anderson *et al.* [57] can simulate cognitive and motor skill transfer across speed perturbations of the same environment. Our main research hypothesis is that humans only recalibrate the statistical knowledge pertaining to the tuning of parameters that have been directly perturbed by contextual and environmental changes. To address this research question, we made a simpler variant of the *Space Fortress* video game, which we called *Auto Orbit*. The goal of the game is to learn to alternate turns and shots with the right timing to progressively blow and burst a balloon as many times as possible. Preliminary work on *Auto Orbit* has shown that a number of motor learning measures including the entropy of keypress sequences, shot timing variability (defined as the logarithmic CV of inter-shot intervals), shot periodicity and shot regularity (defined with the autocorrelation function) were linear predictors of the game score over the course of skill acquisition, and could account for 79% of the variance in game score at a slow speed and 88% of the variance in game score at faster speeds [67]. Importantly, learning the right timing in *Auto Orbit* was dependent on feedback from balloon resets and deflations, which indicated whether the player was playing too fast or too slowly.

We have three main hypotheses pertaining to cognitive and motor skill transfer in the *Auto Orbit* video game. Our first hypothesis is that players can transfer their learned skills across speed perturbations by recalibrating shot timing skills across speeds. Our second hypothesis is that skill acquisition can be accounted for by motor learning signatures including an increased consistency in motor behavior: namely, we expect that skill acquisition will be characterized by a decrease in entropy of keypress action chunks and a decrease in shot timing variability (log CV). Our third and final hypothesis is that inter-individual differences in skill level can be

accounted for by the four previously introduced measures of motor learning, namely the entropy of keypress sequences, shot timing variability, shot periodicity and shot regularity. The first two hypotheses are predictions of our ACT-R model. While the model did not address inter-individual differences, the third hypothesis seemed a plausible extension to individual differences.

## Methods

### *Auto Orbit* video game

The *Auto Orbit* game involves a spaceship that is flying in an orbit at a fixed speed clockwise around a balloon (circle-shaped target) centered on the screen (see Fig 1). The goal is for the player to adjust the ship's aim and periodically shoot missiles within an assigned firing interval. Successful shots result in the balloon being inflated by 1/10 of its full size and are marked by a brief electronic sound. Once fully inflated, the player needs to execute a quick double shot shorter than 250 ms to burst the balloon and complete a game cycle. Balloon bursts were each rewarded by a fixed number of points that corresponded to the current game speed. Each time a missile missed the balloon, the player was penalized by a loss of 2 points regardless of speed. In this experiment, we divided games into a series of game cycles that all started with a "balloon respawn" event and ended with a "balloon burst" event.

A spaceship is flying in a clockwise orbit around a balloon. The player needs to learn to adjust the spaceship's aim and shoot periodically to inflate the balloon and burst it with a double shot. Reprinted from [67] under a CC BY license, with permission from Terrence C. Stewart, original copyright 2020.

The player could execute one of three actions: rotate clockwise by 15 degrees ("D" key), rotate counterclockwise by 15 degrees ("A" key), and launch a missile ("L" key). Throughout the game, subjects learned to pace their shots to maintain a firing rate with an inter-shot interval (ISI) between an imposed lower and upper bound. When shots were faster than the firing interval's lower bound, they resulted in a balloon "reset" marked by the balloon popping on

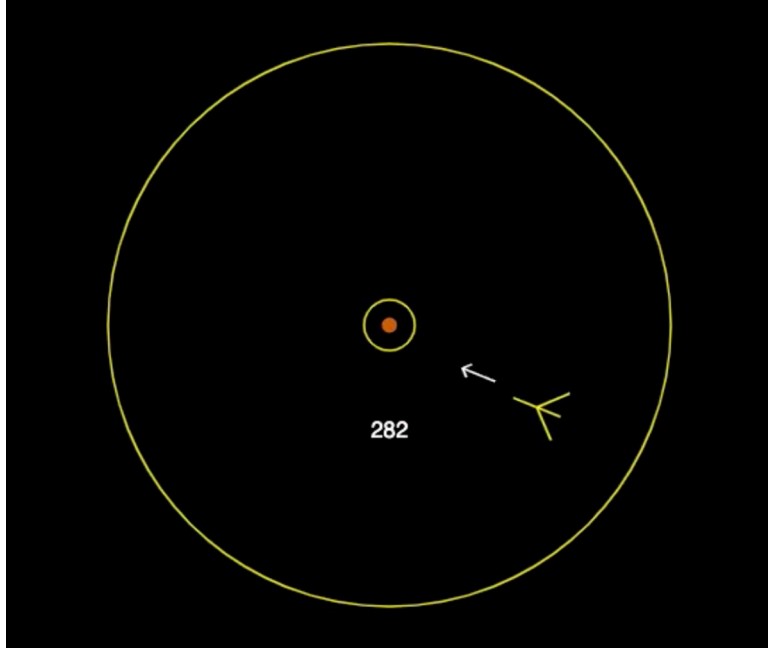

**Fig 1. Illustration of the *Auto Orbit* video game interface.**

Table 1. Description of the two game speeds: High and low speeds.

| Game Speed | Speed Multiplier | Resets | Deflations | Points |
|---|---|---|---|---|
| High | 1.0 | 500 ms | 1,200 ms | 200 / burst |
| Low | 0.5 | 1,000 ms | 2,400 ms | 300 / burst |

the screen with a popping sound. Conversely, when shots were slower than the firing interval's upper bound, they resulted in balloon "deflations" characterized by the balloon progressively shrinking at a constant rate of 0.5% of the balloon's full size (or 0.09 pixels) per game tick of ~16 ms (or 60 updates per second) in addition to a standard balloon deflation sound.

As the ship flew around the balloon, subjects needed to turn their ship to keep it aimed at the balloon and this involved mainly clockwise ship rotations to compensate for the orbiting motion. To add some noise in the video game, random ship rotations of 60 to 120 degrees were scheduled with 1/3 probability at the beginning of every game cycle. This rotation happened at a random time after the cycle began, uniformly chosen over the interval 1 to 4 seconds. Subjects had 4 s. after each ship rotation to adjust the ship's aim and resume firing before the balloon started deflating. An illustration of the *Auto Orbit* interface is depicted in Fig 1. One can play the *Auto Orbit* video game by clicking on the following link: http://andersonlab.net/orbit/signin.html. On this website, the low speed is referred to as the "slow" speed and the high speed is referred to as the "medium" speed.

## Experimental design

The experiment consisted of a total of 15 games that were each 3 minutes in duration (45 minutes in total). Subjects played games at one of two speeds: the high and low game speeds (see Table 1). At high speed, the ship's orbital speed was 0.5 pixels per game tick (~16 ms), completing a full rotation around the balloon roughly took 25 s., and the missile speed was 5 pixels per game tick. Players needed to fire missiles within the [500 ms– 1200 ms] interval and balloon bursts were each rewarded by a game score increase of 200 points.

At low speed, all aspects of the game were reduced by a ratio of 0.5 which included the timing of shots (see Table 1). Monetary compensation was proportional to participants' average game score. Skill was operationalized as the total number of points earned per game and was independently assessed across speeds. Because there were less frequent opportunities to earn points at the slower speed, balloon bursts led to a reward of 200 points at high speed and 300 points at low speed (see Table 1).

To assess transfer across speeds, we utilized an ABA design in which A and B correspond to two different speeds. Since there are 2 game speeds in total, we investigated $2^2 = 4$ conditions (see Table 2). The "HHH" and "LLL" control conditions only involved one game speed whereas the "HLH" and "LHL" transfer conditions involved speed transfer from high speed to low speed, and transfer from low speed to high speed respectively. In each condition, players

Table 2. The four transfer ABA conditions.

| | | Speed 2 | |
|---|---|---|---|
| | | Low | High |
| Speed 1 | Low | LLL | LHL |
| | High | HLH | HHH |

LLL, low-low-low; LHL, low-high-low; HLH, high-low-high; HHH, high-high-high. LHL and HLH are the experimental Transfer conditions. LLL and HHH are the control No Transfer conditions.

completed a first set of 5 games at the first speed (Start phase), then transferred to the second speed for another set of 5 games (Middle phase), and finally transferred back to the original game speed for the 5 last games (Final phase). A log file was recorded with 16-ms temporal resolution in each game. A number of game events were recorded at every time stamp which included "hold-key", "release-key", "random rotation", "vulnerability reset", "vulnerability increase", "missile fired", "balloon respawn" and "balloon burst" events.

### Ethics statement

All experimental procedures were approved by Carnegie Mellon University's Institutional Review Board (IRB). Human subjects that took part in this study all provided consent online before starting the experiment.

### Human subjects

Human subjects were recruited on the Amazon Mechanical Turk (mTurk) online platform. Eighty-two individuals took part in the experiment. Out of those, two participants scored less than 100 points per game (on average) and were excluded from all analyses. We are thus reporting data from 80 participants randomly assigned to each of four transfer game speed conditions (20 participants per game speed). Participants were aged 20 to 40 years-old ($M = 30.0$, $SD = 5.1$). Twenty-one were female, 58 were male, and one individual reported identifying with another gender. In terms of handedness, there were 73 right-handed individuals, 4 left-handed individuals, and 3 reported being ambidextrous. Subjects earned a base pay of $4 for completing the experiment, in addition to a bonus which was proportional to their game score (in points) as specified on Table 1. On average, participants earned a bonus of $5.69.

### Description of key ACT-R components

ACT-R models were all inspired from Anderson *et al.* [57]'s model parameterization. We refer the reader to S1 Table for a list of all key model parameters. The following model components were particularly important in *Auto Orbit*:

a. *Operator retrieval*. Instructions were encoded in ACT-R's declarative memory as a set of proceduralized operators (see S1 Text for the specific instructions). Operators were stored as chunks and retrieved when their activation level was higher than all competing chunks' activation level [56]. Note that, as a simplification, base-level learning mechanism was switched off in these model simulations. Our rationale is that the Auto Orbit task is an inherently procedural task in which the only type of chunks that is encoded in declarative memory is operators, which represent the instructions that participants read prior to starting the experiment. These few chunks are accessed constantly and would show very little variation in base-level learning and hence this factor would have little effect on game performance.

b. *Game state buffer*. Because of the rapid visual changes in the video game environment, the model received information about the state of the game in a game state buffer as part of the visual module. The game state buffer was updated roughly every 16 ms (1/60 s) and included information about the ship position (x-y coordinates), the ship's angle relative to the balloon, the ship velocity, the balloon vulnerability level (inflation level), as well as information about resets and deflations.

c. *Motor productions.* Motor actions were executed under ACT-R's parameterized sub-symbolic constraints. On average, it took the model 50 ms to prepare and execute motor movements. Both hands had separate execution stages, and the timing of motor actions was not completely deterministic due to added noise.

d. *Temporal module.* Action timing was informed by ACT-R's temporal module [63] which could account for effects of memory contamination and time pressure in temporal interval estimation tasks [64,65]. Pulses were accumulated and monitored in a buffer through a pacemaker-accumulator internal clock model [68,69]. Each time a request was sent to the temporal buffer, the pulse count was reset to 0, a chunk was inserted into the temporal buffer with count 0, and the pulse count then started incrementing. Pulses were accumulated in the form of time ticks, where $t_{n+1} = 1.1 \times t_n + noise$ ($\mu = 0$, $\sigma = 0.0165 \times t_n$) where $t_n$ corresponds to the n[th] time tick being accumulated, and where noise was generated using a logistic distribution.

e. *Production compilation.* Production compilation is a major component of the ACT-R learning process, which accounts for much of skill acquisition. It produces a gradual transition from motor actions that are guided by the retrieval of operators to direct responses that are situation-specific [56]. When two productions fire in sequence, they can be combined into a novel production, which is then added to procedural memory [70]. Because the new compiled production is more efficient than the original one, it will gradually acquire a greater utility and be chosen instead of the original. Concretely, utilities reflect the time required to perform an action. Thus, production compilation relies on a utility learning mechanism that determines which productions may apply in cases in which more than one production matches the given initial conditions. When building up the utility of a new compiled production, the utility of its first parent is used before firing for the first time. Production utilities are updated using the following difference learning rule (see Eq 1).

$$U_i(n) = U_i(n-1) + \alpha[R_i(n) - U_i(n-1)] \tag{1}$$

where $U_i(n)$ corresponds to the n[th] update of the i[th] production utility, $R_i(n)$ corresponds to the reward at the n[th] update, $U_i(n-1)$ corresponds to the i[th] production utility at the n-1[th] update, and the parameter $\alpha$ is the learning rate (see S1 Table). When they apply, compiled productions can be combined with other productions eventually collapsing a long sequence of productions that interprets an operator into a single direct production.

f. *Controller module.* The Controller module estimated and refined the range of key model parameters in individual trackers. For each tracker, the Controller module started with greater tolerance and narrowed the tolerance over time, so that values could be progressively tuned. The main tracker of interest monitored shot timing. Participants started with no information about how long the appropriate firing interval should be. Therefore, the model started with a wide range of 10 to 30 time ticks (175 to 1,809 milliseconds). Two other important parameters that must be learned concerned the aim of the ship (i.e., angular orientation relative to the balloon center). The first one represents the ideal aim offset: The ship is always moving and its aim relative to the balloon is changing. The model must learn how far its aim should be from directly at the ship so that when it does fire it will have moved to be directly aimed at the ship. Specifically, the model searches an offset range from -18 to 0 degrees. The second tracker parameter pertaining to the ship's aim represents how far a shot should be from directly at the balloon and still hit the balloon. The model searches a width, or angle tolerance, from 5 to 15 degrees (two times the width to the left, and one time the width to the right of the ship's target aim). At any given time *t*, the Controller

module assigns a probability to each tracker for selection. The probability distribution for this selection is governed by the *softmax* equation (see Eq 2)

$$P(S, t) = \frac{e^{V(S)/T(t)}}{\int_{min}^{max} e^{V(x)/T(t)}} \tag{2}$$

where S is a particular value within the range, V(S) is an ACT-R specified function of S, and T(t) is a temperature function that governs the standard deviation of V(S). The denominator is a scaling factor to make the probabilities sum to 1. Following the argument in Lucas *et al.* [71] for simple functions, Anderson *et al.* [57] chose a simple quadratic function V(S) to describe payoff as a function of the control setting S. One advantage of quadratic functions is that they have a maximum, which facilitates the identification and updating of good values. With respect to the temperature function, T(t) decreases as the model starts selecting the option that seems best, making the quadratic function more peaked over time. Specifically, the temperature decreased as follows (see Eq 3).

$$T(t) = \frac{A}{1 + B \cdot t} \tag{3}$$

where *A* is the initial temperature set to 1.0, *B* is a scaling factor set to 1/180, and *t* is expressed in seconds. Unlike the earlier model from Anderson *et al* [57], the Controller module included a decay mechanism where selected values were progressively forgotten according to an exponential function (i.e., decay) [72]. Specific values for each key parameter can be found in S1 Table.

## ACT-R mechanism of skill acquisition

All ACT-R models presented in this paper shared the same core mechanism of skill learning. Fig 2 provides an illustration of the ACT-R operators that were used to build up skill. As mentioned earlier, acquiring skill in *Auto Orbit* involved learning to alternate turns and shots in a way that would maintain a shooting rate within the firing interval. Sequentially, ACT-R

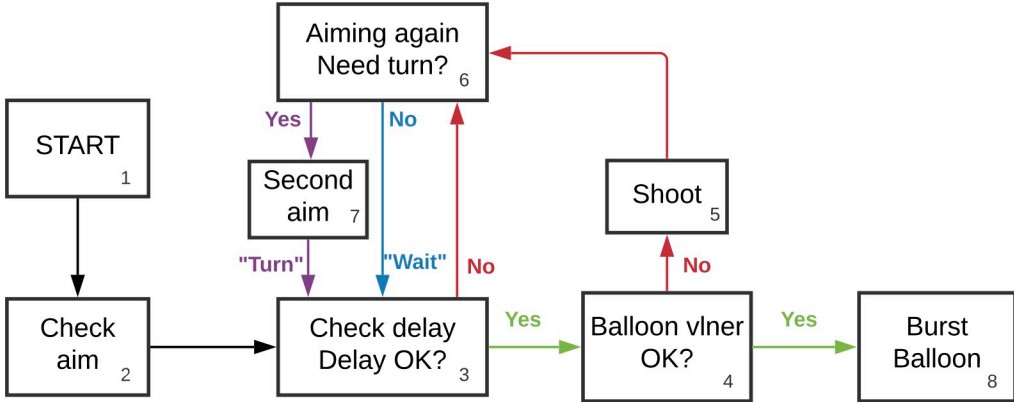

**Fig 2. Description of ACT-R operators in *Auto Orbit*.** The diagram is displaying the order in which operators (action rules based on the instructions) were retrieved from the start (balloon respawn–box 1) to the end (balloon burst–box 8) of a game cycle. The "check aim" operator (box 2) checked whether the spaceship was correctly aimed at the balloon. The "check delay" operator (box 3) assessed whether the temporal module time tracker was greater than a given threshold. The "balloon vlner" operator (box 4) assessed whether the vulnerability of the balloon was high enough for ACT-R to execute a double shot (box 8). If it was too low, then ACT-R executed a single shot (box 5). The "aiming again" operator (box 6) assessed whether the ship's aim was too narrow with respect to the center of the balloon and required an additional turn (box 7) (Note: This diagram was created in Lucidchart, www.lucidchart.com).

models first checked the spaceship's aim with respect to the balloon and adjusted it based on information from two trackers that monitored the aim offset and the aim width in the Controller module (box 2). Second, the temporal module kept track of time and waited for long enough before launching a missile at the balloon (box 3). Waiting times were based on shot timing information stored in a single tracker in the Controller module. Third, depending on the balloon vulnerability (box 4), the model could either execute a simple shot if the vulnerability was too low (box 5), or a double shot if it was high enough (box 8). Fourth, in cases in which the model was considering an additional turn, it first assessed its aim with respect to the balloon based on a threshold that could vary from -10 to 0 degrees (box 6). Note that the threshold was tracked and progressively adjusted by the Controller module in a separate tracker. If the aim was judged wide enough with respect to the threshold, then ACT-R would select a "short" option which would skip the turn. On the other hand, if the aim was judged too narrow with respect to the threshold, then it would fire a "long" option which executed an additional turn (box 7; see Fig 2).

### ACT-R manipulations

We investigated two manipulations of ACT-R parameters. First, to assess the role of the shot timing tracker in skill transfer after game speed perturbations, we compared two ACT-R models that either requested or did not request a new shot timing tracker at the new game speed (Middle phase). Second, to assess the role of temperature reset in control tuning, we compared two ACT-R models that either reset or did not reset the temperature when requesting a second shot timing tracker in the Middle phase. We thus compared a total of three different ACT-R models, which were run 100 times in each of the four conditions (400 model runs in total). Note that each of the three models were initialized with the exact same parameters, which reduced the variability among the 100 models run in a condition.

### Experimental measures

We evaluated a total of eight experimental measures of skill acquisition divided into two sets. The first set included measures of success of game play (performance): these included the game score as a measure of skill (see Table 1), reset counts per game to indicate the number of shots that were too fast (faster than the firing interval lower bound), deflation counts per game to indicate the number of shots that were too slow (slower than the firing interval upper bound), and miss counts per game to indicate the number of missiles that failed to hit the balloon target (measured as the number of fired missiles minus the number of balloon hits per game). Of note, to ensure average results were not dominated by a few extreme values, resets and deflations were capped at 100 per game and misses were capped at 200 per game.

The second set consisted of four motor learning measures inspired from previous research: the keypresses' sequential entropy, shot timing variability, shot periodicity, and shot regularity. All motor learning measures were computed across game cycles ("balloon respawn" to "balloon burst" events) without random rotations for every player and every game. In addition, rows without autocorrelation non-zero positive peaks were filtered out. This filtering removed 7.17% of the human data, 6.68% of the ACT-R model data with one shot timing tracker, 0.27% of the ACT-R model data with two shot timing trackers and one temperature reset, and 0.28% of the ACT-R model data with two shot timing trackers and two temperature resets.

a. *Entropy* The entropy measured keypresses' sequential variability in *Auto Orbit*. We focused on the relative frequency of keypress triples, which were shown to be an optimal chunking length in recent research [73]. With three keys ('L': shoot, 'A': turn counterclockwise, 'D':

turn clockwise) there are $3^3 = 27$ triples. We computed keypress triple probabilities per game by using a non-overlapping counting method (Python count() function) and Laplace smoothing for each keypress triple in all game cycles. We used the Shannon entropy measure, which quantifies unpredictability of information content in a probability distribution [74]. Shannon entropy's formula is shown below (see Eq 4)

$$H(X) = -\sum_{i=1}^{27} p_i \cdot \log_2 p_i \tag{4}$$

where X refers to a game number and $p_i$ refers to the probability of the i[th] triple. This entropy measure could vary from 0 (only 1 triple throughout) to 4.75 (all triples equally likely).

b. *Shot timing variability.* In order to measure shot timing variability, we first extracted ISIs in milliseconds within game cycles. For each game of every player, we computed the coefficient of variation (CV) which is defined as the standard deviation divided by the mean of the ISIs, consistent with previous research [40,42]. An average CV of the ISIs was computed across game cycles within each player's game. We carried out data transformation on CV and calculated its logarithm as this measure was shown to linearly predict the game score during skill acquisition in *Auto Orbit* [67].

c. *Shot periodicity and regularity.* Shot periodicity and shot regularity measures were computed based on the shots autocorrelation function within game cycles. For each game cycle, we first re-preprocessed players' log files such that we would get a single discrete time series of shot events, where individual entries corresponded to successive game ticks of 16 ms. At every tick, a 1 referred to the shot key being held, and a 0 referred to the shot key *not* being held. We could then compute the shots' time series autocorrelation in terms of a correlation coefficient [75] at a particular lag $l$ (see Eq 5) by running the 'acf' function from the statsmodels time series analysis ('tsa') library in Python [76].

$$r_l = \frac{\sum_{i=1}^{N-l}(x_i - \bar{x})(x_{i+l} - \bar{x})}{\sum_{i=1}^{N}(x_i - \bar{x})^2} \tag{5}$$

In the above equation, each $r_l$ corresponds to the shots discrete time series with $N$ elements correlated with a shift of itself by a lag $l$, which concretely measures the degree to which 1s and 0s are aligned across time series after a lag $l$. In this experiment, each autocorrelation function was comprised of 125 time lags of 16 ms. We averaged the autocorrelation function across game cycles without random rotations for each player's individual games. As a result, each player had a total of 15 autocorrelation functions corresponding to each of the 15 games. Fig 3 displays an example of a game autocorrelation function in a subject. Positive peaks in this function reflect lags at which the fire keys tended to be pressed.

We used each game autocorrelation function to extract our two measures of interest: shot periodicity and shot regularity. To do so, we identified the first non-zero lag positive peak of the autocorrelation function indicative of the maximum alignment with the shifted shots time-series. As a way to filter out autocorrelation positive peaks that are due to noise, we only considered peaks whose amplitude was greater than 0.02. In terms of experimental measures, *shot periodicity* was defined as the peak's lag (in ms) at which fires tended to be pressed, and *shot*

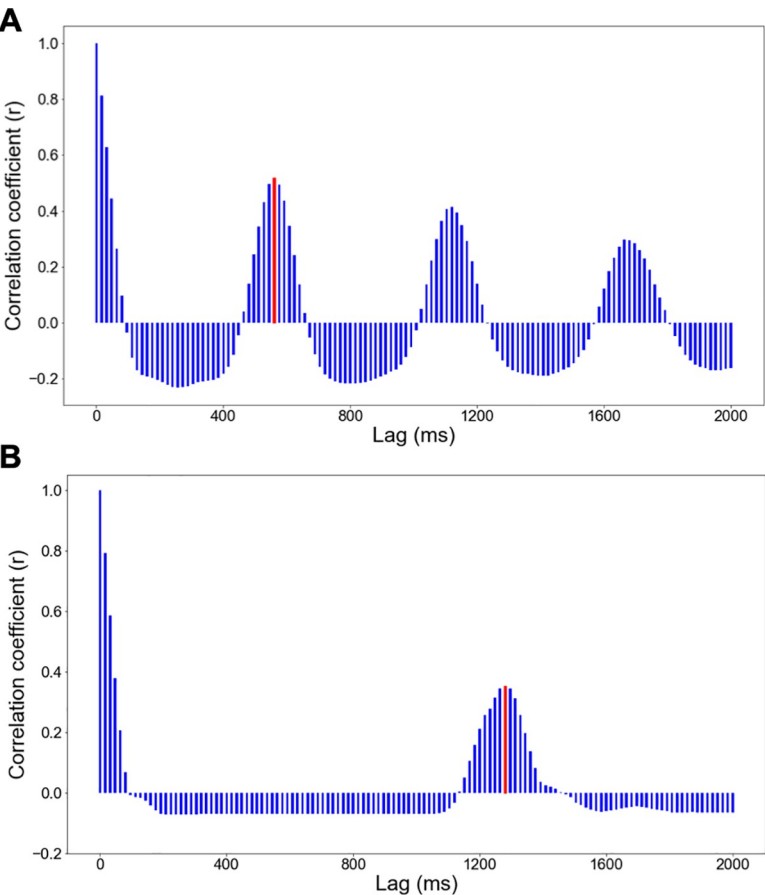

**Fig 3. Illustration of the autocorrelation function.** (A) Example of a participant's autocorrelation function at high speed with a shot periodicity of 560 ms. (B) Example of a participant's autocorrelation function at low speed with a shot periodicity of 1280 ms. On both bar graphs, the red bar indicates the first non-zero positive peak, which is used to compute the player's *shot periodicity* (x-coordinate) and the player's *shot regularity* (y-coordinate).

*regularity* was defined as the height of this peak (correlation coefficient) and reflected how regularly keypresses occurred at this lag. On Fig 3, the first non-zero autocorrelation positive peak has been identified with a red bar for each of the two game speeds.

## Data evaluation

We first explored the evolution of each of the eight experimental measures over the 15 games. We plotted humans' and models' game means over the course of the experiment and provided an estimate of humans' standard deviation to indicate their variability levels. Since all models were initialized with the same parameters, there was little variation across model runs. To assess model fit for each of the three ACT-R models, we first computed the root mean squared error (RMSE) between models' and humans' game means for each experimental measure. RMSEs provided an estimate of how much models deviated from the human means, and were computed on vectorized human and model game means across conditions. Since there were 4 game speed conditions and 15 games, we compared 4x15 = 60 game means across models and humans. In order to compare ACT-R models and determine which fit best, we computed the Bayesian Information Criterion (BIC) for each of the three ACT-R models. Specifically, BIC

was computed as a function of the residual sum of squares as follows (see Eq 6)

$$BIC = n \times \log \frac{RSS}{n} + k \times \log(n) \qquad (6)$$

where $n$ is the number of observations (60 game means), $RSS$ is the residual sum of squares and $k$ is the number of estimated parameters. In the context of this experiment, note that all ACT-R models were initialized with the same number of ACT-R parameters (see S1 Table). We thus elected to focus the BIC comparisons on $\times log \frac{RSS}{n}$, effectively setting $k$ to 0 for all model BICs.

To assess skill transfer across models and humans, we compared the Middle phase (games 6 to 10) Transfer and No Transfer average game scores with respect to the Start phase (games 1 to 5) average game score at that same speed. Specifically, we computed a measure of % transfer using the following formulae inspired by past research (see Eqs 7 and 8; [49,53])

$$T_{Low}(\%) = \frac{Points_{HLH_{6-10}} - Points_{Low_{1-5}}}{Points_{LLL_{6-10}} - Points_{Low_{1-5}}} \times 100 \qquad (7)$$

$$T_{High}(\%) = \frac{Points_{LHL_{6-10}} - Points_{High_{1-5}}}{Points_{HHH_{6-10}} - Points_{High_{1-5}}} \times 100 \qquad (8)$$

where $Low_{1-5}$ corresponds to the LHL and LLL conditions at games 1 through 5 and $High_{1-5}$ corresponds to the HLH and HHH conditions at games 1 through 5. A custom bootstrapping function was run in R to provide an estimate of variability in transfer. We first computed the average game scores across games 1 through 5 for the Start phase, and across games 6 through 10 for the Middle phase in humans and ACT-R models separately. We then computed 10,000 bootstrapped measures of transfer for low and high speeds in humans and models. For each bootstrap, we sampled with replacement from the original averaged data set and computed the % transfer according to Eq 7 for high to low transfer, and according to Eq 8 for low to high transfer.

Focusing on the first two phases, we looked at effects of speed on measures of motor learning and effects of practice on motor learning. Averaging the data within phases, there is a 2x2 set of conditions crossing speed and practice. The difference between the Start Phase and the Middle Phase, averaged over speed, provides a practice measure. The difference between high and low speeds, averaged over phases, provides the effect of speed. Since subjects switch speed in Transfer conditions but not in the No Transfer conditions, we performed two separate analyses for the No Transfer vs. Transfer conditions. Due to missing data across all games in either of the Start or Middle phase, we filtered out the data from 4 out of 80 human individuals. Since the data did not meet the normality and homoskedasticity assumptions, we performed non-parametric Wilcoxon tests ("wilcox.test" function in R) to assess effects of phase and speed. For both the No Transfer and Transfer conditions, we estimated differences between low and high speeds with an independent-samples Wilcoxon rank sum test with a 95% confidence interval. In terms of skill acquisition, we estimated potential differences between the Start and Middle phases of the No Transfer conditions with a two-tailed paired-samples Wilcoxon signed rank test with 95% confidence interval. With regards to phase effects related to transfer, we compared the Middle phases of the Transfer and No Transfer conditions with an independent-samples Wilcoxon rank sum test to assess whether transferring to a new speed led to better or worse motor performance.

Finally, we assessed inter-individual differences with multi-level correlations and a linear mixed-effects model. The main goal was to determine whether our measures of motor learning

could account for differences in skill level in *Auto Orbit*. To do so, we first averaged individuals' game scores and motor learning across games from the Final phase (last 5 games). To assess overall correlation levels across the four conditions, we computed multi-level correlations with the "correlation" function from the R "correlation" package [77]. We then fit a linear mixed-effects model on the squared average game scores in the Final phase. The reason for performing a squared data transformation is that many individuals' game scores were close to ceiling, thus yielding a skewed distribution.

We set up the four motor learning variables as fixed effects and added a random intercept to account for variability in game scores across conditions. Specifically, we estimated the four $\beta$ coefficients corresponding to each motor learning variable in the following equation: $Y_{Score}^2 = \beta_1 X_{Entropy} + \beta_2 X_{logCV} + \beta_3 X_{Periodicity} + \beta_4 X_{Regularity} + Z_{GameSpeed}$. Note that the model also estimated a random intercept $Z$ for each condition (HHH, HLH, LHL, and LLL). In R, we imported and ran functions from the lme4 [78] and lmerTest [79] packages to fit the linear mixed-effects model. The model was written as $lmer$($Score^2$ ~ $Entropy$ + $logCV$ + $Periodicity$ + $Regularity$ + (1|$GameSpeed$)). The restricted maximum likelihood (REML) technique was employed to fit each model and we obtained $p$-values for each $\beta$ estimate. A QQ-plot of the model's residuals did not reveal any obvious deviations from normality, which was confirmed by a Shapiro-Wilk normality test ($W$ (80) = 0.975, $p$ = 0.13). Finally, visual inspection of the residual plots did not reveal any obvious deviations from homoskedasticity. We provided the 95% confidence interval (CI) for each estimate, which we computed using bootstrapping (resampling with replacement). We simulated 1,000 resampling experiments and extracted the coefficients corresponding to each of the 4 fixed factors ("fixef") using the R function bootMer. We then extracted the CI lower and upper bound with the nonparametric R function "boot. ci".

## Results

Fig 4 shows subject performance on the four measures of game success. In developing an analysis of these results and others the section will be divided into three parts. The first part will explore how the results can be understood in terms of ACT-R models and the second two parts will further explore effects of skill transfer and inter-individual differences respectively. In terms of modeling, we compared three ACT-R models of skill transfer in *Auto Orbit*. There were two manipulations of interest: 1) Tracker manipulation, 2) Temperature manipulation.

### 1. ACT-R manipulation results

We first assessed ACT-R models' ability to transfer their skills following game speed perturbations in the Middle phase of the experiment. Specifically, we focused our investigation on the transfer of motor timing skills and compared ACT-R model trackers that either requested or did not request a new shot timing after a speed switch (Middle Phase). The first ACT-R model preserved the original trackers from the Start phase for the entirety of the experiment regardless of speed transfer. This meant that when the game speed changed, the model had to adjust its estimate of V(S) in response to the new feedback, basically unlearning the old control setting and learning a new control setting. Thus, this model counted on the decay of old experiences in the Controller and the buildup of new experiences to adjust to the change in speed [72]. We refer to this model as the "1 tracker model".

The second ACT-R model started learning over again when it recognized there was a speed change. It suspended the current shot timing tracker and made a request for a new shot timing tracker that would start learning V(S) without the weight of the prior experience. This model only started a fresh estimation of V(S) but kept the reduced value of the temperature T(t).

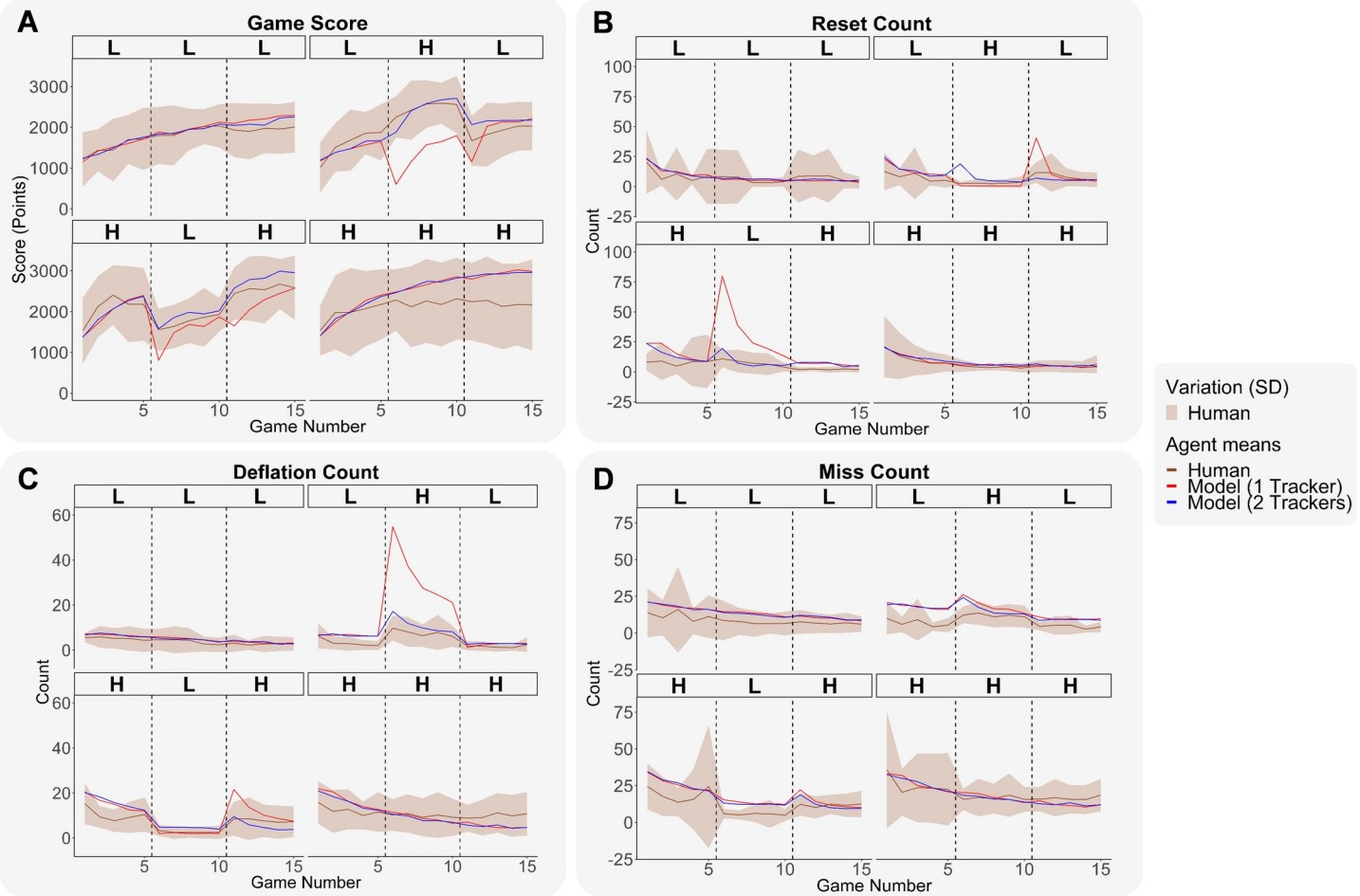

**Fig 4. Human and ACT-R model performance results with one vs. two trackers across the four conditions.** (A) Game score over the 15 games. (B) Reset count over the 15 games. (C) Deflation count over the 15 games. (D) Miss count over the 15 games. Across all plots, human means are shown with brown continuous lines and human standard deviations are indicated with a light brown shaded area. ACT-R model means with one tracker are shown with red continuous lines. ACT-R model means with two trackers are shown with blue continuous lines. Dashed vertical lines indicate phase switches.

When re-exposed to the initial speed at the start of the Final phase (game 11), the model simply reactivated the first shot timing tracker and completed the experiment this way. The request for a second shot timing tracker did not immediately happen in the transfer conditions' Middle phase. The model explored the environment for one cycle to get a sense of a reasonable firing range. It then made the new shot timing tracker request to the Controller module with the new estimated firing range. Note that there was no tracker reset in the control conditions (LLL and HHH) since the model did not detect any speed changes in the Middle phase. We refer to this model as the "2 trackers model".

Fig 4 shows the fit of the two models and the first two rows of Table 3 provides measures of fit for these two models. We refer the reader to S1 and S2 Figs for a visual description of humans and ACT-R model data spread within games. Humans and ACT-R models had similar learning trajectories characterized by fast improvements early on that progressively reached a plateau within speeds. ACT-R models with two trackers had game scores close to humans and were within a standard deviation off the human means, whereas ACT-R models with only one tracker (first row on Table 3) showed a considerable disadvantage characterized by a major drop in their game score at the sixth game (transfer to new speed) and at the eleventh

**Table 3. Goodness of fit in terms of RMSE and BIC and model comparison across performance measures.**

| | Model description | | Model fit—Performance | | | |
|---|---|---|---|---|---|---|
| | Trackers | Temperature Resets | Points | Resets | Deflations | Misses |
| **RMSE** | 1 | 1 | 479 | 11.63 | 8.73 | 6.68 |
| | 2 | 1 | 304 | 4.73 | 3.32 | 6.30 |
| | 2 | 2 | 312 | 5.09 | 3.72 | 6.51 |
| **BIC** | 1 | 1 | 741 | 294 | 260 | 228 |
| | 2 | 1 | 686 | 186 | 144 | 221 |
| | 2 | 2 | 689 | 195 | 158 | 225 |

game (transfer to original speed; see Fig 4A). One surprising finding was that subjects from the HHH condition converged towards an asymptote earlier than ACT-R models, and humans' game score asymptote in that condition tended to be lower than models' game score asymptote (see Fig 4A). We believe that this result is due to some HHH subjects' poorer skill levels at high speed. Indeed, HHH subjects had a significantly lower game score ($Mdn$ = 1844 points) than HLH subjects ($Mdn$ = 2128 points) in the Start phase (i.e., first 5 games at high speed), $z$ = -2.17, $p < .05$. In terms of model fits, resetting the shot timing tracker at the new speed led to improvements both in terms of RMSE and BIC (see Table 3; see Tables A and B in S2 Table).

Fig 4B–4D respectively present the evolution of reset counts, deflation counts, and miss counts over the games. As individuals learned to shoot within their assigned firing interval, resets and deflations progressively decreased. Similarly, as players learned to correctly adjust the ship's aim with respect to the balloon, there was a progressive decrease in miss counts over the games. Deflations and misses tended to decrease when transferring from high speed to low speed and increase when transferring from low speed to high speed, and an opposite trend was weakly present in resets. In addition, we noticed that the model with one tracker did not recalibrate its shot timing knowledge in the first game of the second speed, which was characterized by a dramatic peak of resets at game 6 when transferring from high to low speed, and by a peak of deflations at game 6 when transferring from low to high speed. In terms of model fits, this resulted in RMSE and BIC improvements in the model with two trackers (see Table 3; see Tables A and B in S2 Table). Overall, while models' performance tended to be within the human range, there was more variability in human game score measures than in ACT-R model game score measures (see S1 Fig).

We next investigated the four measures of motor learning over the games across models with one vs. two trackers (see Fig 5). Once more, one can notice that the ACT-R model with two trackers better fit humans' motor learning trend than the ACT-R model with one tracker in terms of RMSE and BIC (see two first rows on Table 4). In addition, ACT-R models tended to have lower sequential variability (entropy) than humans, particularly in the LLL condition (see Fig 5A). Both humans and ACT-R models quickly adjusted their shot periodicity to the new game speed in games 6 and 11 of the Transfer conditions (see Fig 5B). ACT-R models tended to have lower shot timing variability levels than humans' average shot timing variability levels as suggested our measure of log CV ISI (see Fig 5C). The ACT-R models' sequential and shot timing variability levels were close to the better performing human subjects, suggesting that these models reflect the behavior of the more consistent individuals. With regards to shot regularity, humans and models' regularity levels progressively increased over the course of skill acquisition (see Fig 5D). Although models generally had lower variance than humans, their distributions generally fell within the human range (see S2 Fig).

Overall, the main result from tracker manipulations was that the ACT-R model with two trackers generally better fit humans than the model with one tracker in terms of performance

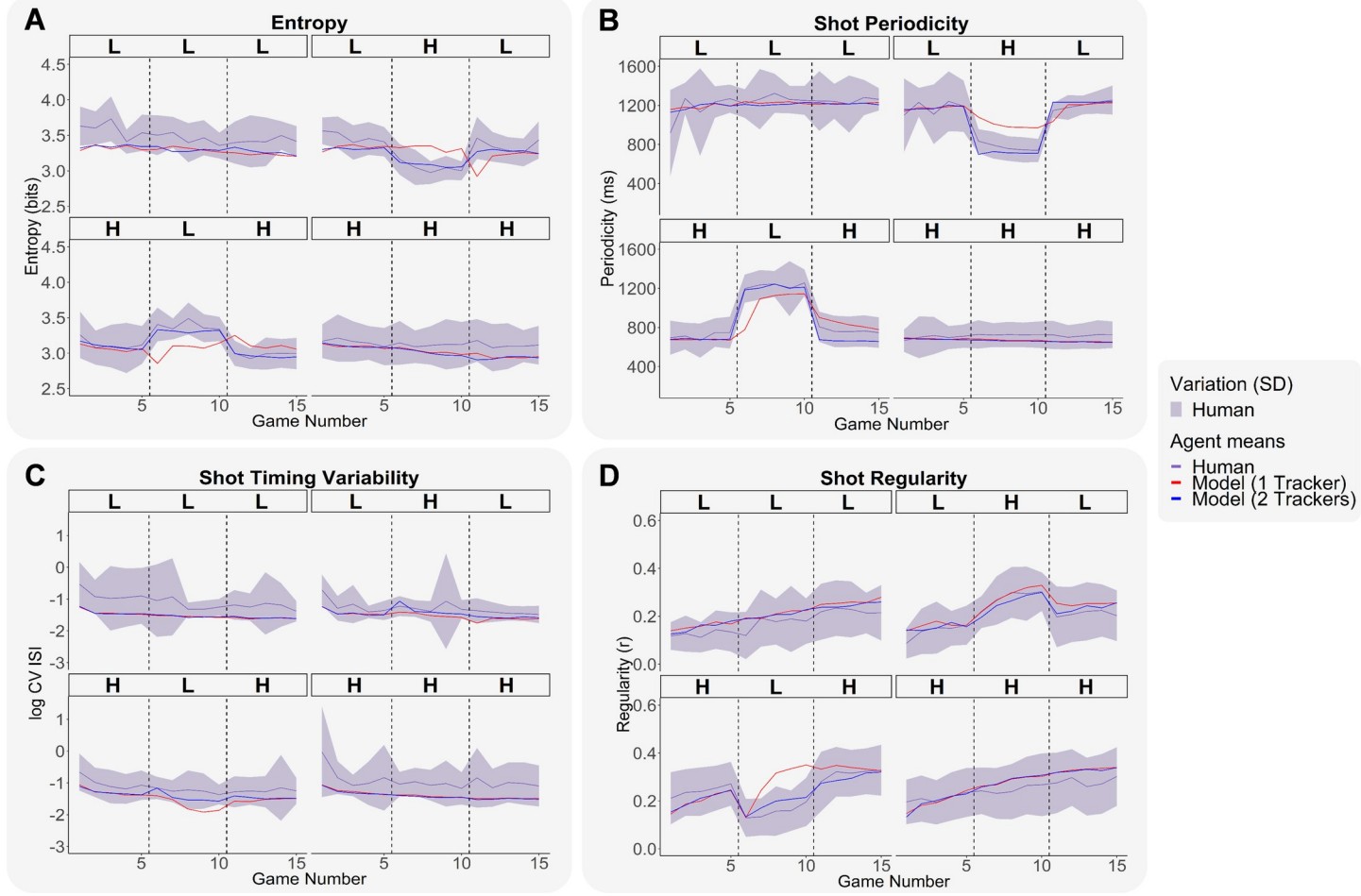

**Fig 5. Human and ACT-R model motor learning measures with one vs. two trackers across the four conditions.** (A) Entropy over the 15 games. (B) Shot periodicity over the 15 games. (C) Shot timing variability over the 15 games. (D) Shot regularity levels over the 15 games. Human means are shown with purple continuous lines and human standard deviations are indicated with a light purple shaded area. ACT-R model means with one tracker are shown with red continuous lines. ACT-R model means with two trackers are shown with blue continuous lines. Dashed vertical lines indicate phase switches.

(see Table 3) and motor learning (see Table 4). This model kept the reduced temperature of the old tracker. We next inquired whether resetting the temperature at the new speed led to further model fit improvements (i.e., temperature manipulation). We thus built a third ACT-R model to assess whether a reduced temperature in the new tracker was detrimental to model performance when transferring to a new speed. The new model reinitialized the same

**Table 4. Goodness of fit in terms of RMSE and BIC and model comparison across motor learning measures.**

| | Model description | | Model fit–Motor Learning | | | |
|---|---|---|---|---|---|---|
| | Trackers | Temperature Resets | Entropy | Periodicity | Regularity | Log CV ISI |
| **RMSE** | 1 | 1 | 0.21 | 107 | 0.05 | 0.40 |
| | 2 | 1 | 0.14 | 66 | 0.03 | 0.37 |
| | 2 | 2 | 0.14 | 66 | 0.04 | 0.37 |
| **BIC** | 1 | 1 | -188 | 561 | -353 | -109 |
| | 2 | 1 | -233 | 503 | -406 | -118 |
| | 2 | 2 | -235 | 502 | -392 | -118 |

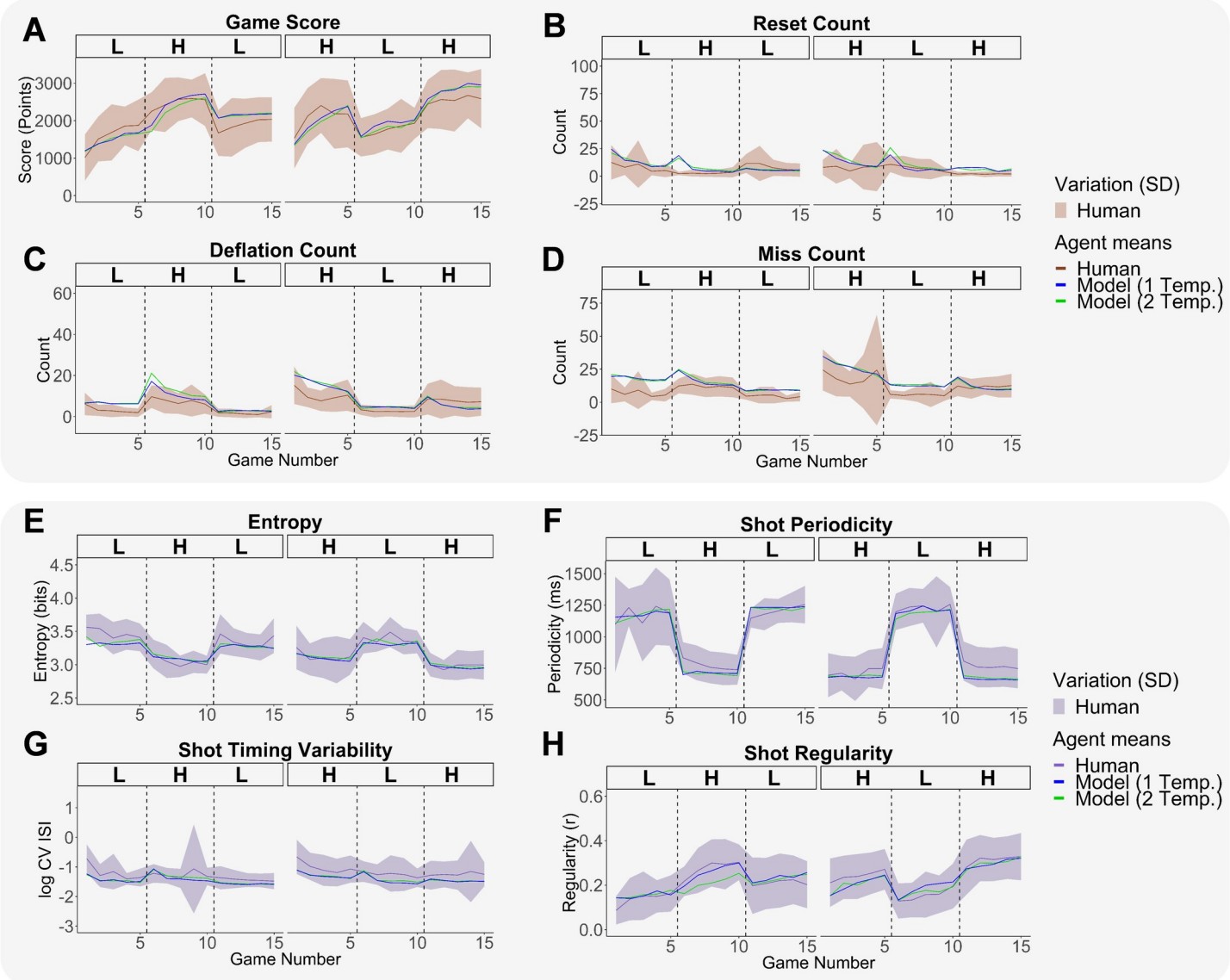

**Fig 6. Performance and motor learning measures across humans and ACT-R models with and without a new temperature reset in their second shot timing tracker.** (A) Game score over the 15 games. (B) Reset count over the 15 games. (C) Deflation count over the 15 games. (D) Miss count over the 15 games. (E) Entropy over the 15 games. (F) Shot periodicity over the 15 games. (G) Shot timing variability over the 15 games. (H) Shot regularity over the 15 games. Across all plots, human performance means and standard deviations are shown in brown while human motor learning means and standard deviations are shown in purple. ACT-R model means with one temperature reset are shown with blue continuous lines. ACT-R model means with two temperature resets (1 per speed) are shown with green continuous lines. Dashed vertical lines indicate phase switches.

temperature parameter to its original value when exposed to the second speed. No other aspects of the model were modified, and the original shot timing tracker was also reactivated at the beginning of the Final phase (game 11) meaning that it also utilized 2 different trackers. Thus, this model will show greater initial variability in the second phase as it explores a wider range of values.

Fig 6 displays a comparison between ACT-R models with an extra temperature reset (third row of Tables 3 and 4) and without an extra temperature reset (second row). Each experimental measure's evolution is plotted over the fifteen games in the two Transfer conditions. In

terms of the RMSE measures in Table 3, the model with one temperature reset was marginally better than the model with two temperature resets (see Fig 6A and Table 3). We notice that there were slightly fewer resets and deflations during the transfer to a new speed at game 6 in the model with one temperature (see Fig 6B & 6C), but similar miss counts across models with and without a new temperature (see Fig 6D). In terms of motor learning (see Table 4), sequential variability levels (entropy) and shot timing variability (log CV ISI) were not improved with a temperature update. That being said, preserving the old temperature did provide an advantage in terms of shot regularity, particularly when transferring from low to high speed (see Fig 6H & Table 4). Finally, there was a minor advantage for using a second temperature in terms of shot periodicity (see Fig 6F & Table 4). Our main conclusion from the previous model fit analyses (RMSE & BIC) was that the best fitting ACT-R model requested a new shot timing tracker at the new speed but preserved the temperature from the Start phase.

## 2. Skill transfer across humans and models

We then investigated skill transfer across humans and ACT-R models by applying Eqs 7 and 8 to game scores. We focused our analyses on the model with two trackers and one temperature. Humans and ACT-R models both had remarkably high levels of transfer across speeds that were all greater than 50% (transfer to high speed: Human $Mdn$ = 131%; ACT-R $Mdn$ = 68%; transfer to low speed: Human $Mdn$ = 76%; ACT-R $Mdn$ = 82%). We computed bootstrapped estimates of the variability in transfer across individuals and model runs. ACT-R models had lower variability than humans and their spread mostly fell within humans' interquartile range at low speed (see Fig 7). With regards to high speed, we notice that humans had transfer levels that were higher at high speed than at low speed. Humans' high speed transfer estimate (exceeding 100%; see Fig 7) likely reflects the overall poor learning of subjects in the HHH condition (reducing the denominator in the calculation of transfer for high speed). On the other hand, ACT-R transfer levels remained comparably similar at high and low speed.

Fig 8 displays the effects of speed and phase on measures of motor performance, separately for the No Transfer and Transfer conditions (see Table 5 for statistics). With respect to effects of speed, the results were consistent for the two conditions: subjects were slower at low speed (the required result), less regular, had higher entropy, and showed no significant effect on log CV ISI. As to effects of phase, within-subject comparisons in the No Transfer conditions also yielded consistent results: Regularity increased from the Start to Middle phase, entropy decreased, and log CV ISI decreased. The ACT-R model with two trackers show effects consistent with the significant effects of both speed and phase (see S3 Fig).

## 3. Inter-individual differences

Finally, we assessed whether the four motor learning measures could account for inter-individual differences in terms of game scores. To do so, we predicted a squared transformation of average human game scores across the five final games (Final phase) and computed multi-level correlations across individuals' average game scores and their entropy, shot timing variability, shot periodicity and shot regularity. Table 6 provides the results from a linear mixed-effects model and multi-level correlations where each of the four conditions (HHH, HLH, LHL, LLL) was represented by a random intercept. The main result was that the entropy and shot timing variability (log CV ISI) measures were both predictive of individuals' skill levels but shot periodicity and shot regularity were not.

This result was consistent with multi-level correlations, which showed stronger correlations for entropy and log CV ISI than shot periodicity and shot regularity. Decreases in entropy and decreases in log CV ISI were overall correlated with higher game scores, regardless of the

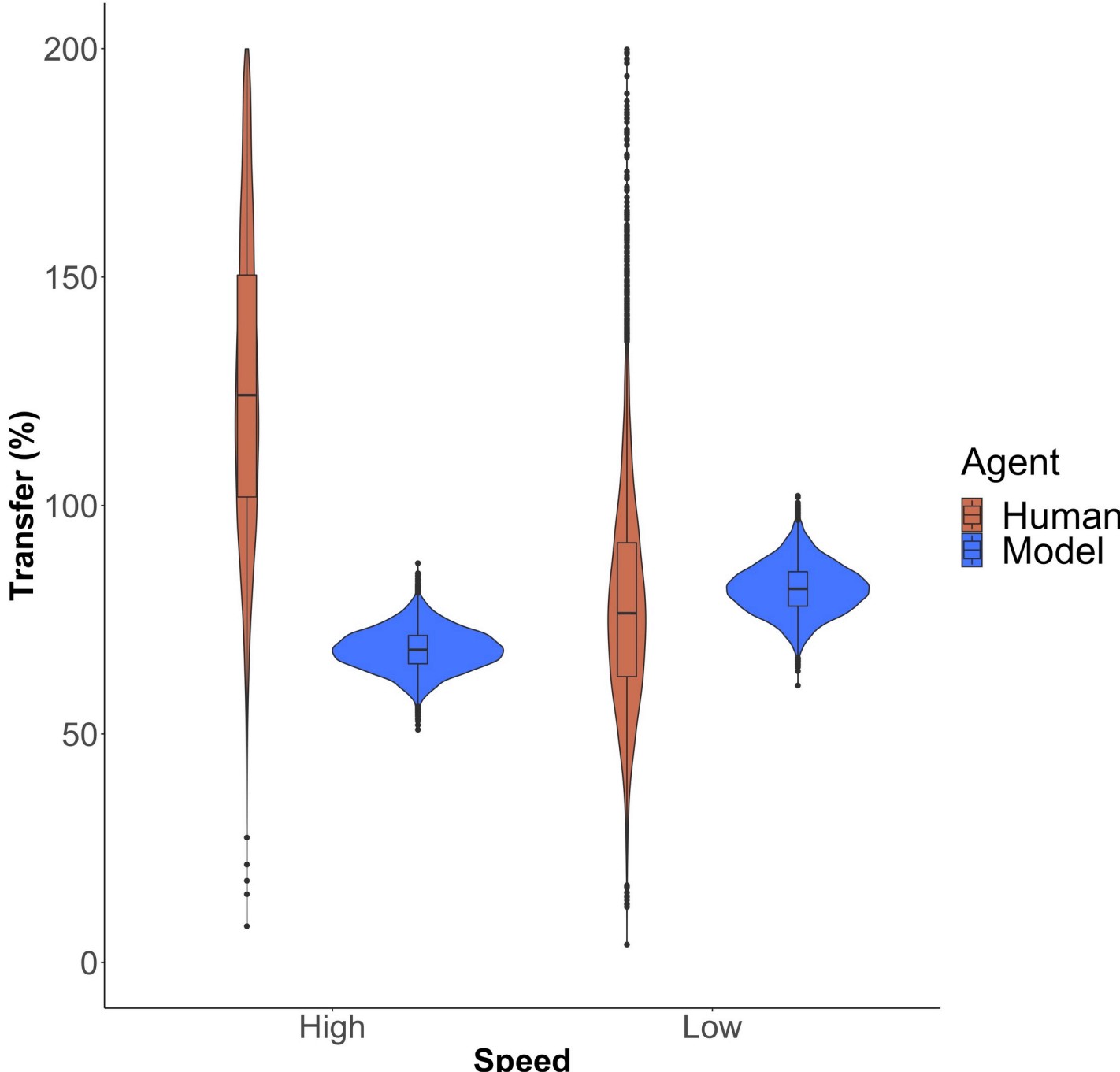

**Fig 7. Human and model bootstrapped transfer levels at high and low speeds.** The violin plot displays the frequency distribution of bootstrapped transfer levels (%) in humans (brown) and in the ACT-R model with two trackers and one temperature (blue). For each speed (high vs. low), a boxplot indicates the transfer median, first quartile and third quartile.

condition the participant was in. There was no clear predictive effect of shot regularity and shot periodicity on skill levels since their confidence interval lower and upper bound were of opposite signs. Correlations between entropy and game scores within conditions are shown on Fig 9A, and correlations between log CV ISI and skill levels within conditions are shown on

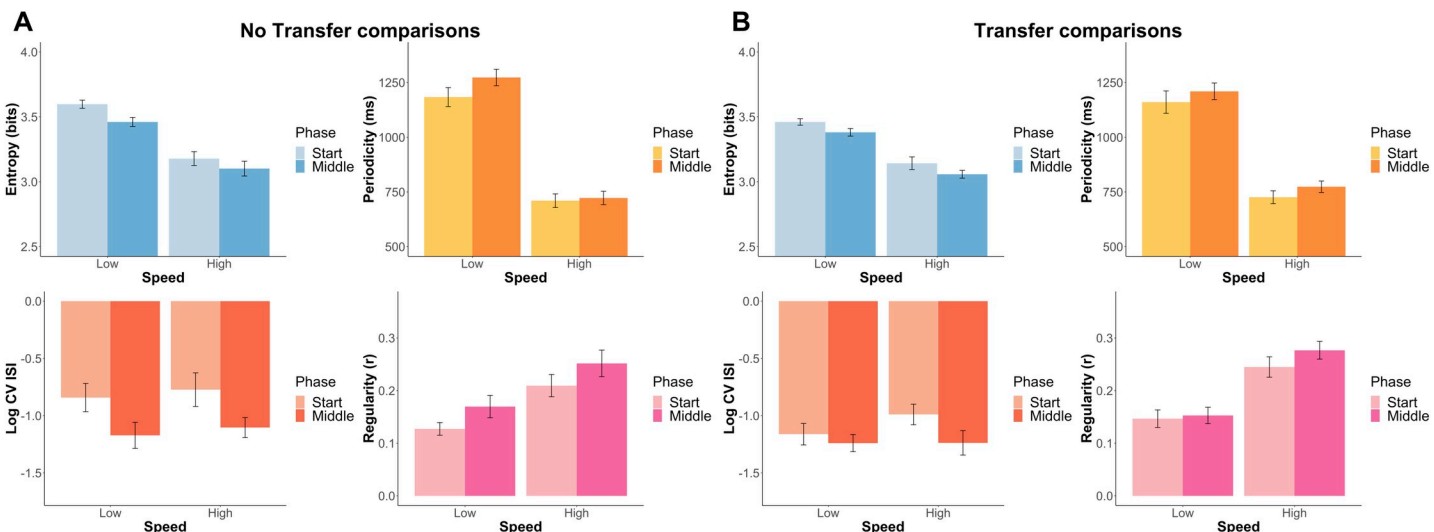

**Fig 8. Comparison of human measures of motor learning across the Start phase (5 first games) and the Middle phase (5 middle games) of *Auto Orbit*.** (A) No Transfer phase comparison of human entropy (blue), shot periodicity (yellow), shot timing variability (red), and shot regularity (pink) in the LLL (low speed only) and HHH (high speed only) conditions. (B) Transfer phase comparison of human entropy (blue), shot periodicity (yellow), shot timing variability (red), and shot regularity (pink) in the LHL (Low-Start & High-Middle) and HLH (High-Start & Low-Middle) conditions. The bar graph indicates means and standard errors of the means within speeds and phases.

Fig 9B. Correlations between periodicity and skill levels are shown on S4A Fig, and correlations between regularity and skill levels are shown on S4B Fig.

## Discussion

We employed a simple video game, *Auto Orbit*, to further advance our understanding of cognitive and motor skill transfer across speeds. We investigated two speeds that were proportional to one another in all aspects by a ratio of 0.5. Out of the four conditions of interest, two had no transfer and acted as our control conditions (one game speed only), and two others were our experimental conditions where agents transferred from one speed to the other at game 6, and then transferred back to the original speed at game 11. To assess cognitive and motor skill transfer, we utilized a set of four experimental measures related to performance: game score, reset counts, deflation counts and miss counts, and a second set of four experimental measures related to motor learning in *Auto Orbit*: keypress sequential variability, shot

**Table 5. Transfer and no transfer comparisons across speeds and phases.**

| | | | Motor learning tests | | | | | | | |
|---|---|---|---|---|---|---|---|---|---|---|
| | | | Entropy | | Periodicity | | Log CV ISI | | Regularity | |
| | Test | Paired | z | p | z | p | z | p | z | p |
| No Transfer conditions | Speed | No | -6.08 | < .001 | -7.11 | < .001 | -0.89 | 0.37 | -3.86 | < .001 |
| | Phase* | Yes | -3.20 | < .01 | -1.24 | 0.22 | -4.13 | < .001 | -3.58 | < .001 |
| Transfer conditions | Speed | No | -6.39 | < .001 | -6.65 | < .001 | -1.04 | 0.30 | -5.39 | < .001 |
| | Phase* | No | -1.13 | 0.26 | -0.01 | 0.99 | -1.59 | 0.11 | -0.34 | 0.73 |

The tests of Speed compared High vs. Low speeds in the Transfer and No Transfer groups respectively.

*No Transfer tests of Phase compared the Start vs. Middle phases in the No Transfer conditions, whereas Transfer tests of Phase compared the Middle phases of the No Transfer vs. Transfer conditions.

**Table 6. Linear mixed-effects model and multilevel correlations of inter-individual differences in skill.**

| | LMEM | | | Multi-level correlations | | |
|---|---|---|---|---|---|---|
| | B | 95% CI | | r | 95% CI | |
| Entropy | -6510301*** | (-9229947, | -3862927) | -0.63*** | (-0.75, | -0.48) |
| Log CV ISI | -1606117** | (-2703241, | -405376) | -0.62*** | (-0.74, | -0.47) |
| Periodicity | -2743 | (-8964, | 4027) | -0.22* | (-0.42, | 0.00) |
| Regularity | 2791891 | (-1184017, | 7283639) | 0.38*** | (0.17, | 0.55) |
| Adjusted $R^2$ | 0.80 | | | | | |

*** $p < .001$

** $p < .01$

* $p < .05$; $\beta$s refer to linear estimates and $r$ is a multilevel correlation coefficient [77].

timing variability, shot periodicity, and shot regularity. Specifically, learning how to play *Auto Orbit* involved acquiring two main skills: 1) learning the right alternation of keypresses between "turn" and "shoot" actions, 2) learning how to shoot at the right periodicity according to the lower and upper bound of the assigned firing interval. Critical to modeling these results was the single tracker in the Controller module that encoded information pertaining to the shot timing interval.

We investigated a number of questions relevant to how to understand the course of learning and transfer within a computational architecture like ACT-R. The first question was that skill transfer would involve the recalibration of shot timing skills across speeds. This was answered

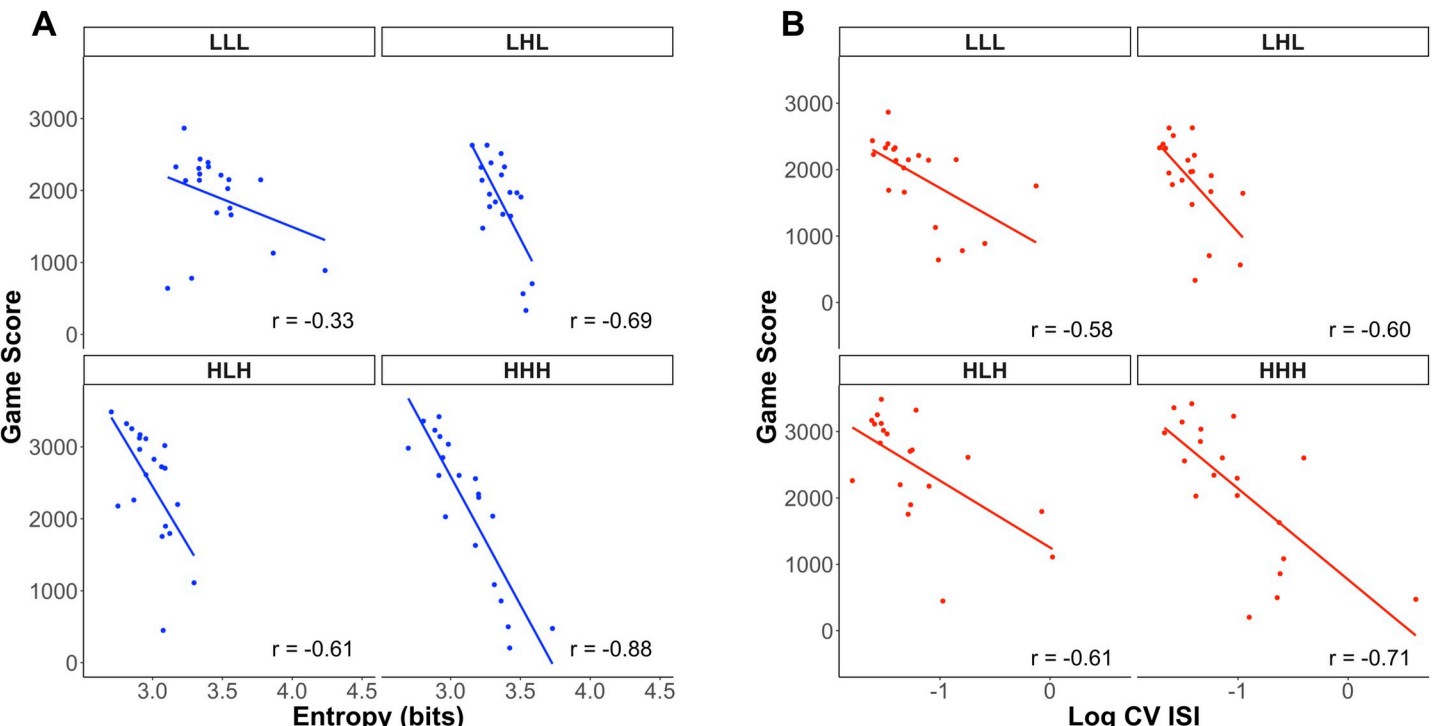

**Fig 9. Correlation between human average game score and human average motor variability in terms of the entropy and shot timing variability in the Final phase.**
(A) Correlation between human average game score and average entropy. (B) Correlation between human average game score and average shot timing variability in terms of the logarithmic CV of the ISIs.

by comparing models with one or two trackers. Here, initializing a second tracker after the speed switch enabled the architecture to match subjects' adaptation to the new speed, thereby avoiding a significant performance drop during transfer. The better performance of the two-tracker model that did not reset the temperature suggests that humans are more confident about their performance after exploring their environment early on and have less of a tendency to continue this exploration in later games. This is consistent with reinforcement learning accounts of motor learning and motor variability [80]. Furthermore, ACT-R model fits demonstrated that humans successfully recalled shot timing statistical information from the Start phase in the Final phase, which indicates that past learning experiences likely leave a trace. While past ACT-R models have often represented the context-dependent learning and adaptation of timing in terms of declarative memory retrieval mechanisms and instance-based learning [64], we showed here that timing skills could also be represented along with other sensorimotor skills in the form of a tracker in ACT-R's Controller module, which bears more similarity to internal models of sensorimotor learning [26].

One limitation of the ACT-R model is that it solely relies on feedback to adapt its behavior. When transferring from speed to speed, ACT-R first needs to explore the range of possible shot timing parameter values to successfully adapt to the new speed. This early exploration was characterized by small increases in number of resets at the beginning of transfer. Humans, however, did not show this peak in resets. One potential explanation is that participants may have been extracting information pertaining to the physical dynamics of the video game in addition to the feedback from resets & deflations to help them transfer across speeds. This interpretation fits with predictions from dynamical systems theories on motor movement and motor timing [81,82].

In terms of skill transfer, we observed that there was remarkable transfer across speeds in *Auto Orbit*. This suggests that humans may be reusing learned information from the initial games to adapt their behavior to speed perturbations of the environment in *Auto Orbit*. In terms of the model, this happened because of the proceduralization of the instructions via production compilation, which gets transferred from speed to speed. This supports the proposal that production compilation acts as a general skill acquisition process within tasks [53] whereas control tuning contributes to an agent's adaptation to specific environments based on feedback and dynamic sensorimotor learning [57].

Interestingly, while human transfer levels matched ACT-R transfer levels at low speed, we noticed that humans had higher transfer levels than ACT-R at high speed and ACT-R transfer levels remained comparably similar at high speed and low speed. There are two ways to interpret this result. On the one hand, one could argue that slow practice may provide an advantage when transferring to faster speeds. In practice, it is common for teachers to ask their students to slow down as a way to optimize skill learning both in music [83,84] and in the martial arts [85]. As it turns out, recent neurorehabilitation studies have started to provide evidence referencing the benefits of slow-down exercises. For instance, it has been shown that musicians with focal hand dystonia (i.e., with involuntary flexion and extension of fingers during music performance) could recover some of their somatosensorial, sensorimotor and muscular skills after slow-down exercise training [86,87]. On the other hand, since subjects from the high-speed only condition performed relatively poorly, it could be the case that such high transfer levels at high speed in humans are in fact due to inter-individual skill differences across game speed conditions.

The second hypothesis was that skill acquisition would be characterized by motor learning signatures. In addition to the progressive adaptation to speeds in terms of shot periodicity, motor learning consisted of decreased variability in action chunking and timing, and increased shot regularity. To test the reliability of the *Auto Orbit* video game, we first compared shot

periodicity levels across speeds and phases. As expected, individuals' shot periodicity levels were slower at low speed and faster at high speed regardless of transfer. Moreover, we did not find differences across early and middle game phases.

In terms of action chunking, we found that skill acquisition in *Auto Orbit* resulted in lower keypress sequential variability in the Middle phase regardless of transfer. This result suggests that learning how to play a video game requires one to learn successful statistical keypress action patterns that are robust to speed perturbations and control tuning. In the model, a major factor producing this result was production compilation. As the model increased the speed of executing its actions, sequences of shots were less often interrupted by needed turns. This finding is consistent with a body of research suggesting that action chunking may act as a cost-efficient strategy to reduce computational complexity and progressively leads to more efficient movements [38,88,89]. In terms of speed differences, we found that there was more sequential variability at low speed than at high speed. We believe that this result is due to the greater number of degrees of freedom in terms of possible combinations of action sequences at low speed.

With regards to timing, we found that later game phases were characterized by lower timing variability (log CV ISI). Like the model, subjects progressively learned the video game's timing constraints within speeds and adapted the pacing of their shots accordingly. Interestingly, we did not find significant differences of speed. These findings support past research on timing in motor skill acquisition [39,90], and suggest that decreases in timing variability may be a general marker of skill acquisition regardless of speed and circumstantial tempi [39].

Unlike previous motor timing skill learning studies, we distinguished between shot timing *variability* (defined with the coefficient of variation) and shot *regularity* (defined with the auto-correlation function). In addition to decreases in timing variability, we found that skill acquisition was marked by progressively higher shot regularity levels over time. However, one main difference between the timing variability and regularity measures is that shot regularity levels were found to be higher at high speed and lower at low speed which supports past sensorimotor synchronization findings [39,40,91–93], whereas shot timing variability levels were found to be the same across speeds. One potential explanation is that CV adjusts individuals' variability in timing according to keypresses' average inter-press-intervals [40,42]. The shot regularity measure does not have such adjustment and assesses shot regularity based on the shots' auto-correlation in the game cycle time series instead of computing a general statistic based on the overall mean and standard deviation of the ISIs.

The final hypothesis was that inter-individual differences in skill acquisition can be accounted for by measures of motor learning. Linear mixed-effects model analyses and multi-level correlations revealed that motor variability in terms of action sequencing and timing was the main predictor of inter-individual differences in skill levels regardless of transfer. This result is consistent with a growing body of research [94–98] and demonstrates that the acquisition of timing skills in psychomotor tasks can be predicted by agents' motor variability levels [99]. In terms of motor control, a decrease in variability during motor skill acquisition may reflect feedback control [25,99] and error correction [100]. Shot periodicity and regularity, on the other hand, were found to be time-dependent motor learning predictors of skill acquisition in *Auto Orbit* [67] but did not reliably predict inter-individual differences in skill levels.

This ACT-R model explains most of skill learning in terms of proceduralization of the instructions (i.e., production compilation) and progressive tuning of cognitive and motor skills via the Controller module. Learning to tune a skill critically depends on sensorimotor interactions between executed motor actions (e.g., precisely timed key presses) and their resulting visual feedback (i.e., game state updates), which is in line with the notion of internal model [26,30,57]. However, other more perceptual types of learning may have played a role since

video game experts are usually characterized by shorter reaction times than video game novices, potentially reflective of higher visual and attentional skills [101–103]. In *Auto Orbit*, perceptual skills may have been at play during the early stages of the game when players learn to act upon visual and auditory feedback (e.g., balloon resets/deflations and their corresponding sounds). In terms of ACT-R modeling, higher perceptual skills may be partly captured by faster rates of production compilation, particularly in early games. Nevertheless, such interpretation should be taken with caution as it is hard to disentangle pure perceptual learning from sensorimotor learning.

In terms of the Controller module, it is still unclear how feedback-related learning is instantiated in the brain. To reiterate, control tuning is believed to simulate the experience-dependent process whereby humans adjust their behavior through trial and error based on feedback they receive from the environment [57]. From a cognitive neuroscience perspective, researchers have long hypothesized that the *cerebellum* may act as an internal model to predict the sensory consequences of motor actions, and has sometimes been modeled in terms of a Smith predictor [27,30,104,105]. Further research on motor learning has shown that the cerebellum leverages a sensory prediction error as teaching signal to implement supervised learning computations in the brain [27,29,31]. In terms of feedback, recent findings have revealed that patients with cerebellar brain damage were still able to rely on (lagged) visual feedback in motor learning [104]. This suggests that feedback learning does not only recruit the cerebellum, but also involves a broader network of brain regions interacting with one another. Thus, the cerebellum might play a key role in control tuning. Future research is needed to shed light on the precise neurobiological mechanisms that underlie feedback-related computations in the Controller module.

In sum, we have found that skill transfer across speed perturbations of the environment required the recalibration of action timing skills. We showed that skill transfer was characterized by progressive action chunking and production compilation which facilitated transfer. In *Auto Orbit*, acquiring skill involved a progressive decrease in motor variability, and skill transfer across speeds was marked by shot periodicity adjustments and progressively more regular shots. Finally, we found that highly skilled players tended to be more consistent in their keypress patterns and in the timing of their shots. Further work will explore effects of speed and will more precisely uncover the neurobiological mechanisms of control tuning in cognitive and motor skill transfer.

## Supporting information

**S1 Fig. Distribution of performance measures in humans and ACT-R models over the games.** (A) Game score distribution over the 15 games across agents. (B) Reset count distribution over the 15 games across agents. (C) Miss count distribution over the 15 games across agents. (D) Deflation count distribution over the 15 games across agents. Boxplots indicate the median, 1st quartile and 3rd quartile. Across all plots, humans are shown in brown and ACT-R models are shown in blue. Performance measures solely include data from the ACT-R model with two trackers and one temperature. **** $p < .0001$; *** $p < .001$; ** $p < .01$; * $p < .05$. (TIF)

**S2 Fig. Distribution of motor learning measures in humans and ACT-R models over the games.** (A) Entropy distribution over the 15 games across agents. (B) Shot periodicity distribution over the 15 games across agents. (C) Shot timing variability distribution over the 15 games across agents. (D) Shot regularity distribution over the 15 games across agents. Boxplots indicate the median, 1st and 3rd quartiles. Across all plots, humans are shown in brown and ACT-R models are shown in blue. Motor learning measures solely include data from the

ACT-R model with two trackers and one temperature. **** $p < .0001$; *** $p < .001$; ** $p < .01$; * $p < .05$.
(TIF)

**S3 Fig. Comparison of ACT-R measures of motor learning across the Start phase (5 first games) and the Middle phase (5 middle games).** (A) No Transfer phase comparison of ACT-R entropy (blue), shot periodicity (yellow), shot timing variability (red), and shot regularity (pink) in the LLL (low speed only) and HHH (high speed only) conditions. (B) Transfer phase comparison of ACT-R entropy (blue), shot periodicity (yellow), shot timing variability (red), and shot regularity (pink) in the LHL (Low-Start & High-Middle) and HLH (High-Start & Low-Middle) conditions. The bar graph indicates means and standard errors of the mean within speeds and phases. Note that this figure only includes data from the ACT-R model with two trackers and one temperature reset.
(TIF)

**S4 Fig. Correlations between human average game score and average shot periodicity and regularity in the Final phase.** (A) Correlation between human average game score and average shot periodicity. (B) Correlation between human average game score and average shot regularity.
(TIF)

**S1 Table. Table with key ACT-R parameters sorted across function.**
(PDF)

**S2 Table. ACT-R model fits across experimental measures in each of the four conditions.**
(PDF)

**S1 Text. Instructions provided to participants prior to the *Auto Orbit* experiment.**
(PDF)

# Acknowledgments

We thank the International Conference on Cognitive Modeling (ICCM) conference chairs for granting us the permission to reuse a figure from a preliminary conference proceeding. We are grateful for Dan Bothell's technical assistance and help with the implementation of the ACT-R models. Finally, we would like to thank Caitlin Tenison for helpful comments on the draft.

# Author Contributions

**Conceptualization:** Pierre Giovanni Gianferrara, John Robert Anderson.

**Data curation:** Pierre Giovanni Gianferrara.

**Formal analysis:** Pierre Giovanni Gianferrara.

**Funding acquisition:** John Robert Anderson.

**Investigation:** Shawn Betts.

**Methodology:** Pierre Giovanni Gianferrara, John Robert Anderson.

**Software:** Shawn Betts.

**Supervision:** John Robert Anderson.

**Writing – original draft:** Pierre Giovanni Gianferrara.

**Writing – review & editing:** Pierre Giovanni Gianferrara, John Robert Anderson.

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
