## [Decision Letter · Decision Letter 0]

5 Jul 2021

PONE-D-21-05223

Cognitive & motor skill transfer across speeds: a video game study

PLOS ONE

Dear Dr. Gianferrara,

Thank you for submitting your manuscript to PLOS ONE. After careful consideration, we feel that it has merit but does not fully meet PLOS ONE’s publication criteria as it currently stands. Therefore, we invite you to submit a revised version of the manuscript that addresses the points raised during the review process.

Both reviewers are quite positive, but do have suggestions that you should follow. I suspect you will be capable of adapting the paper appropriately for your revision. Perhaps the most critical suggestion is a more full justification for the new control module. The paper should clearly explain the value added compared to the prior similar work that does not use this module.

We look forward to receiving your revised manuscript.

Kind regards,

Micah B. Goldwater, Ph.D

Academic Editor

PLOS ONE

Journal Requirements:

2. We noted in your submission details that a portion of your manuscript may have been presented or published elsewhere.

"Human data from the LLL (low speed only) and HHH (high speed only) game speed conditions were introduced in a preliminary report at the 18th International Conference on Cognitive Modelling (ICCM) (see ICCM 2020 conference proceedings). However, note that the ACT-R model that is introduced in this paper is new and much improved over the preliminary model from the ICCM paper.

The authors believe that this submission does not constitute dual publication because this manuscript explores new modeling ideas for dealing with transfer. The original ICCM conference paper did not address transfer and the preliminary ACT-R model could not model skill transfer across speeds."

Please clarify whether this conference proceeding or publication was peer-reviewed and formally published. If this work was previously peer-reviewed and published, in the cover letter please provide the reason that this work does not constitute dual publication and should be included in the current manuscript.

3. We note that Figures in your submission contain copyrighted images. All PLOS content is published under the Creative Commons Attribution License (CC BY 4.0), which means that the manuscript, images, and Supporting Information files will be freely available online, and any third party is permitted to access, download, copy, distribute, and use these materials in any way, even commercially, with proper attribution. For more information, see our copyright guidelines: http://journals.plos.org/plosone/s/licenses-and-copyright.

3.1.         You may seek permission from the original copyright holder of Figures to publish the content specifically under the CC BY 4.0 license.

3.2.    If you are unable to obtain permission from the original copyright holder to publish these figures under the CC BY 4.0 license or if the copyright holder’s requirements are incompatible with the CC BY 4.0 license, please either i) remove the figure or ii) supply a replacement figure that complies with the CC BY 4.0 license. Please check copyright information on all replacement figures and update the figure caption with source information. If applicable, please specify in the figure caption text when a figure is similar but not identical to the original image and is therefore for illustrative purposes only.

Reviewers' comments:

Reviewer's Responses to Questions

**Comments to the Author**

1. Is the manuscript technically sound, and do the data support the conclusions?

Reviewer #1: Yes

Reviewer #2: Yes

2. Has the statistical analysis been performed appropriately and rigorously? 

Reviewer #1: Yes

Reviewer #2: Yes

3. Have the authors made all data underlying the findings in their manuscript fully available?

Reviewer #1: Yes

Reviewer #2: Yes

4. Is the manuscript presented in an intelligible fashion and written in standard English?

Reviewer #1: Yes

Reviewer #2: Yes

5. Review Comments to the Author

Reviewer #1: This is an excellent paper that addresses an important but often ignored aspect of learning, namely the interaction of motor and cognitive learning as well as the implications of transfer. The background given in the paper nicely sets the context of this work within the larger cognitive modeling community. Both the experimental study and the associated modeling seem extremely well done. The writing is also of top quality.

I had just two minor questions as I read through the paper: (1) Along with the interaction of motor and cognition, the visual system also plays an important role in (almost all?) games including this one. The paper could discuss this a bit further, specifically in arguing that the learning effects are more isolated to motor learning, or perhaps how any potential visual "learning" (or increased efficiency) might be at work here. (2) The paper mentioned that base-level learning was switched off in the model, and yet instructions were represented as declarative items. One might have thought that normally base-level learning would be used for increased efficiency over time. Perhaps the paper should explain more about this, especially since, in general, base-level learning would be active for most realistic models of learning so it would be important to understand how the mechanisms here interact with base-level learning.

Reviewer #2: This is a very interesting paper that discusses a detailed model of acquisition and transfer in a challenging video game. The model is able to explain quite some details of human performance and has impressive model fits. I have a few comments to further improved the paper.

The control model is a recent addition to ACT-R, which makes me wonder about the necessity. The experiment shows quite some similarity to Taatgen, van Rijn and Anderson 2007, which did not use a control module. Some elaboration would be nice.

The paper explores three models with various resetting modes after a change in block. In the resetting conditions: is there also a reset in the control conditions (LLL and HHH)? A possible alternative for resetting is a solution in Taatgen and van Rijn (2011, Memory and Cognition), in which declarative chunks that represented time intervals had an additional context slot. A change in context (i.e., a new block) combined with partial matching would dilute but not wipe out past experience from before the context change.

In general, the presentation of the results can be improved, because currently the reader is forced to jump back and forth in the paper. In the method section a large number of measures and methods are explained, the results then provides a huge number of results without interpretation, that is then provided in the discussion. This may be ok for experimental studies, but for a complex model such as this it doesn’t work.

Page 21: the autocorrelation needs a bit more explanation: I cannot interpret line 477 to 484 very well.

Fig 4-6 are hard to read

Table 3: “Model description” is confusing, because earlier they were called model 1 to 3. Better come up with names and stick to that.

Niels Taatgen

6. PLOS authors have the option to publish the peer review history of their article (what does this mean?). If published, this will include your full peer review and any attached files.

Reviewer #1: No

Reviewer #2: **Yes: **Niels Taatgen

---

## [Author Response · Author response to Decision Letter 0]

14 Aug 2021

<Note: the following text can also be found in our rebuttal letter attached online>

Academic editor:

Both reviewers are quite positive, but do have suggestions that you should follow. I suspect you will be capable of adapting the paper appropriately for your revision. 

> We thank the academic editor for these kind comments.

Perhaps the most critical suggestion is a more full justification for the new control module. The paper should clearly explain the value added compared to the prior similar work that does not use this module.

> We see the editor’s concern and have adapted the manuscript accordingly so that it is clear how the Controller module differs from previous ACT-R modeling work.

We have further motivated the use of the Controller module in several parts of the manuscript:

• Introduction: We added a paragraph to clarify how the Controller module differs from instance-based learning methods used in previous modeling work on time interval estimation – lines 164-184.

• Discussion: We added a paragraph to clarify how control tuning relates to other types of perceptual learning – lines 925-938.

As mentioned in our manuscript, we believe that the main value of the Controller module is its ability to simultaneously simulate a range of precise sensorimotor skills at fast speeds, which can’t be done with declarative memory mechanisms alone. To make this point clearer, we further explained how the Controller may be thought of in terms of an internal model between executed motor actions and their resulting visual feedback.

Please review your reference list to ensure that it is complete and correct.

> We have reviewed our reference list and checked to the best of our ability that it is complete and correct. Updated references have been highlighted in the text, and new references have been highlighted in the reference section.

> The abstract, cover, and supporting information were edited to comply with PLOS ONE’s style requirements. We have also checked that our file names follow the journal guidelines. We hopefully have no divergences from the style requirements now.

2. We noted in your submission details that a portion of your manuscript may have been presented or published elsewhere […]. Please clarify whether this conference proceeding or publication was peer-reviewed and formally published. If this work was previously peer-reviewed and published, in the cover letter please provide the reason that this work does not constitute dual publication and should be included in the current manuscript. 

> Our conference paper was peer-reviewed and published in the 18th International Conference on Cognitive Modelling (ICCM)’s proceedings in 2020. See conference proceedings link below: https://iccm-conference.neocities.org/2020/ICCM2020Proceedings.pdf

We have obtained the written permission from the ICCM 2020 conference chair Dr. Terry Stewart to use the figure in question.

We have clarified how this submission does not constitute dual publication in the original cover letter (see quoted text below).

“A 7-page report summarizing these results was peer-reviewed and published in the 18th International Conference on Cognitive Modelling (ICCM)’s conference proceedings. The enclosed manuscript builds upon the findings from the ICCM conference paper through the exploration of a transfer design and new transfer data. Our novel and much improved ACT-R model explores new modeling ideas for dealing with speed transfer by leveraging the recently introduced Controller module (Anderson et al., 2019). Specifically, we show that speed transfer requires the recalibration of motor timing skills and that variability in behavior can account for skill acquisition and inter-individual skill level differences, which fits with the existing literature (Caramiaux et al., 2018; Ramkumar et al., 2016; Shmuelof et al., 2012).”

We believe that the main reason why this submission does not constitute dual publication is that the current manuscript is predominantly concerned with the transfer of skills across game speed conditions. Our original ACT-R model of skill acquisition from ICCM 2020 was not able to successfully simulate human cognitive and motor skill transfer.

In addition, the current manuscript introduces skill transfer and inter-individual skill analyses that were not considered in ICCM 2020.

We have added a note about the ICCM conference proceeding in the “Acknowledgments” section of the manuscript.

3. We note that Figures in your submission contain copyrighted images. […] We require you to either (1) present written permission from the copyright holder to publish these figures specifically under the CC BY 4.0 license, or (2) remove the figures from your submission.

> Fig 1 was indeed previously published as part of the ICCM 2020 conference proceeding. To ensure that we do not violate any copyrights, we asked the ICCM 2020 conference chair Dr. Terry Stewart to sign the PLOS ONE content permission form, which you can find along with our manuscript on the PLOS ONE portal.

As to Fig 2, Lucidchart is the software we used to produce this figure. Lucidchart’s content terms can be found here: https://lucid.co/tos

As stated under section 5 “Your Content”, “5.1 As between the parties, you own all right, title, and interest in and to the Content in your account, including all intellectual property and proprietary rights therein. Except as expressly set forth herein, Lucid acquires no right, title, or interest from you hereunder in or to your Content. “Content” means the data, information, images, and other content that is uploaded to, imported into or created in the subscription Service by the Users, but does not include Statistical Data (as defined herein).” We thus believe that Fig 2 does not infringe any copyrights.

All other figures have been newly generated in Python or R for this journal publication’s purposes.

Reviewers:

Reviewer #1

This is an excellent paper that addresses an important but often ignored aspect of learning, namely the interaction of motor and cognitive learning as well as the implications of transfer. The background given in the paper nicely sets the context of this work within the larger cognitive modeling community. Both the experimental study and the associated modeling seem extremely well done. The writing is also of top quality.

> We thank the reviewer for these kind comments.

I had just two minor questions as I read through the paper: (1) Along with the interaction of motor and cognition, the visual system also plays an important role in (almost all?) games including this one. The paper could discuss this a bit further, specifically in arguing that the learning effects are more isolated to motor learning, or perhaps how any potential visual "learning" (or increased efficiency) might be at work here.

> We agree with the reviewer that the visual system certainly plays a major role in video games including Auto Orbit. We have addressed this comment in two parts of the manuscript:

• In order to clarify what kind of visual information the ACT-R model is receiving during video game simulations, we added a Methods sub-section under “Description of key ACT-R components” which we entitled “Game state buffer” (lines 321-326). Indeed, in Auto Orbit, the ACT-R model receives all its input related to the game environment via the game state buffer, which was integrated into the visual module.

• To address the reviewer’s point about learning effects and motor vs. visual learning, we have added a paragraph in the Discussion to contrast our results and choice of modeling with perceptual learning effects found in other video game studies (e.g., Bediou et al., 2018; Green & Bavelier, 2003; Dye, Green & Bavelier, 2009) - lines 925-938.

(2) The paper mentioned that base-level learning was switched off in the model, and yet instructions were represented as declarative items. One might have thought that normally base-level learning would be used for increased efficiency over time. Perhaps the paper should explain more about this, especially since, in general, base-level learning would be active for most realistic models of learning so it would be important to understand how the mechanisms here interact with base-level learning.

> The reviewer is bringing up an interesting point about the base-level learning mechanism in ACT-R. We have addressed this point in a Methods sub-section entitled “operator retrieval” under “Description of key ACT-R components”.

“Note that, as a simplification, base-level learning mechanism was switched off in these model simulations. Our rationale is that the Auto Orbit task is an inherently procedural task in which the only type of chunks that is encoded in declarative memory is operators, which represent the instructions that participants read prior to starting the experiment. These few chunks are accessed constantly and would show very little variation in base-level learning and hence this factor would have little effect on game performance.” – lines 314-320.

Reviewer #2

This is a very interesting paper that discusses a detailed model of acquisition and transfer in a challenging video game. The model is able to explain quite some details of human performance and has impressive model fits.

> We thank the reviewer for these kind remarks.

The control model is a recent addition to ACT-R, which makes me wonder about the necessity. The experiment shows quite some similarity to Taatgen, van Rijn and Anderson 2007, which did not use a control module. Some elaboration would be nice.

> We agree that the control tuning of motor timing is similar to past ACT-R models of time interval estimation, such as the one introduced by Taatgen, van Rijn & Anderson (2007). To further clarify how the Controller module differs from past work, we have written an additional paragraph in the Introduction which we think addresses the reviewer’s point.

“Though control tuning mechanisms share some similarity with instance-based learning methods [60–62], as introduced in past ACT-R modeling work on prospective time interval estimation [63–65], they also differ from instance-based learning in important ways that are worth noting. Instance-based learning theories typically assume that increases in performance accuracy levels are due to retrieval across a growing pool of experiences stored in declarative memory [60]. Control tuning does not assume that experiences are stored in declarative memory, and thus does not rely on retrieval strategies. Rather, the Controller is more similar to the notion of “internal model”, which has been developed in the motor learning literature [26,30,66], and refers to unconscious mapping of controllable movement properties (e.g., timing, force, direction) to features of the movement (e.g., state, position, velocity) [57]. In video games, one must simultaneously learn a range of sensorimotor skills at very fast speeds (often < 1 s), which are all required for the successful completion of the goal. Though instance-based learning can certainly simulate the learning and adaptability of one such skill (e.g., time interval estimation [63–65]), it unfortunately falls short when the skills at hand involve a stronger motor component and when the number of sensorimotor skills to learn is too large to keep track of with conscious control from working memory [26]. In terms of computational efficiency, retrieving chunks from declarative memory is costly and often too slow for sensorimotor skills that need to be rapidly and precisely executed [57]. In this sense, control tuning is a useful addition to the ACT-R architecture which nicely complements production compilation to efficiently simulate skill acquisition in complex tasks [57].” – lines 164-184.

The paper explores three models with various resetting modes after a change in block. In the resetting conditions: is there also a reset in the control conditions (LLL and HHH)?

> There is no tracker reset in the control conditions (LLL and HHH). We clarified this point in the Results section.

“Note that there was no tracker reset in the control conditions (LLL and HHH) since the model did not detect any speed changes in the Middle phase.” – lines 626-628.

A possible alternative for resetting is a solution in Taatgen and van Rijn (2011, Memory and Cognition), in which declarative chunks that represented time intervals had an additional context slot. A change in context (i.e., a new block) combined with partial matching would dilute but not wipe out past experience from before the context change.

> We addressed this point about resetting alternatives that leverage declarative memory retrieval strategies in a response to an earlier comment – see Introduction lines 164-184.

We also acknowledged other resetting strategies based on declarative chunk representations, as introduced by Taatgen & van Rijn (2011), in a discussion paragraph concerned with shot timing recalibration – lines 839-844.

In general, the presentation of the results can be improved, because currently the reader is forced to jump back and forth in the paper. In the method section a large number of measures and methods are explained, the results then provides a huge number of results without interpretation, that is then provided in the discussion. This may be ok for experimental studies, but for a complex model such as this it doesn’t work.

> We understand that the first version of our manuscript may have been hard to read due to the structure of the Methods & Results sections. We addressed this point in the revised manuscript and reorganized the description of the models accordingly.

• The methods section now only introduces the core ACT-R mechanisms that are used across all models (i.e., key components & operators).

• The description of each of the 3 ACT-R models is now presented in the Results section along with the figures so that the reader can better appreciate the results as they are reading along.

• To help the reader understand the way we structured the results, we divided our results into three different sections:

1. ACT-R manipulation results

2. Skill transfer across humans and models

3. Inter-individual differences

• With regards to ACT-R results, we framed our results in terms of two manipulations: the tracker manipulation and the temperature manipulation. We included a brief discussion of each manipulation after our presentation of the results before moving on to the next set of results.

Page 21: the autocorrelation needs a bit more explanation: I cannot interpret line 477 to 484 very well.

> The description of the autocorrelation has been revised to clarify how the shot periodicity and shot regularity measures were computed – lines 475-501.

Fig 4-6 are hard to read

> We have improved Figures 4-6 in the following ways:

• Results have been chunked with light gray panels so that it is easier to parse the figures & group the results.

• The legend has been updated to clarify what models are depicted on the figures

• Two different colors are now used on Fig 4 vs. Fig 5 to distinguish performance measures (in brown) from motor learning measures (in purple).

• Dashed vertical lines placed between games 5 and 6, and between games 10 and 11 have been added to clarify when phase switches happen during the game.

• The labels have been updated to clarify what game section corresponds to what speed (L vs. H).

• Titles have been added to clarify what the figures are showing.

• The x-axis and y-axis labels are no longer bolded to ensure the reader’s attention can be drawn to the titles first instead of the axes’ labels.

We also updated S1 and S2 Figs accordingly.

Table 3: “Model description” is confusing, because earlier they were called model 1 to 3. Better come up with names and stick to that.

> We agree with the reviewer that the original way of describing the models may have been confusing. To address this point, we revised our manuscript in the following ways:

1. We first introduced the core ACT-R mechanism of skill acquisition (applicable to all models) in the Methods section.

2. We separately introduced each of the three ACT-R models illustrating the tracker & temperature manipulations in the Results section. The models with vs. without tracker reset are introduced first, and the models with vs. without temperature reset are introduced next.

We dropped our numbering descriptive system (i.e., models 1-3), which did not seem to be an optimal way of referring to the models.

Additional Comments

> We have improved the phrasing of parts of the Introduction and Discussion sections to make the exposition of the paper smoother and clearer from the reader’s perspective. All edited sections have been highlighted in yellow for your convenience. We hope that the current improved version of the manuscript will meet the journal requirements.

---

## [Editor Report · Decision Letter 1]

23 Sep 2021

Cognitive & motor skill transfer across speeds: a video game study

PONE-D-21-05223R1

Dear Dr. Gianferrara,

We’re pleased to inform you that your manuscript has been judged scientifically suitable for publication and will be formally accepted for publication once it meets all outstanding technical requirements.

Kind regards,

Micah B. Goldwater, Ph.D

Academic Editor

PLOS ONE
---

## [Editor Report · Acceptance letter]

30 Sep 2021

PONE-D-21-05223R1 

Cognitive & motor skill transfer across speeds: a video game study 

Dear Dr. Gianferrara:

I'm pleased to inform you that your manuscript has been deemed suitable for publication in PLOS ONE. Congratulations! Your manuscript is now with our production department. 

Kind regards, 

on behalf of

Dr. Micah B. Goldwater 

Academic Editor

PLOS ONE